# Small RNAs from mitochondrial genome recombination sites are incorporated into *T. gondii* mitoribosomes

**Sabrina Tetzlaff[1], Arne Hillebrand[1], Nikiforos Drakoulis[1], Zala Gluhic[1], Sascha Maschmann[1], Peter Lyko[2], Susann Wicke[2], Christian Schmitz-Linneweber[1]***

[1]Molecular Genetics, Humboldt University Berlin, Berlin, Germany; [2]Biodiversity and Evolution, Humboldt University Berlin, Berlin, Germany

**Abstract** The mitochondrial genomes of apicomplexans comprise merely three protein-coding genes, alongside a set of thirty to forty genes encoding small RNAs (sRNAs), many of which exhibit homologies to rRNA from *E. coli*. The expression status and integration of these short RNAs into ribosomes remains unclear and direct evidence for active ribosomes within apicomplexan mitochondria is still lacking. In this study, we conducted small RNA sequencing on the apicomplexan *Toxoplasma gondii* to investigate the occurrence and function of mitochondrial sRNAs. To enhance the analysis of sRNA sequencing outcomes, we also re-sequenced the *T. gondii* mitochondrial genome using an improved organelle enrichment protocol and Nanopore sequencing. It has been established previously that the *T. gondii* genome comprises 21 sequence blocks that undergo recombination among themselves but that their order is not entirely random. The enhanced coverage of the mitochondrial genome allowed us to characterize block combinations at increased resolution. Employing this refined genome for sRNA mapping, we find that many small RNAs originated from the junction sites between protein-coding blocks and rRNA sequence blocks. Surprisingly, such block border sRNAs were incorporated into polysomes together with canonical rRNA fragments and mRNAs. In conclusion, apicomplexan ribosomes are active within polysomes and are indeed assembled through the integration of sRNAs, including previously undetected sRNAs with merged mRNA-rRNA sequences. Our findings lead to the hypothesis that *T. gondii's* block-based genome organization enables the dual utilization of mitochondrial sequences as both messenger RNAs and ribosomal RNAs, potentially establishing a link between the regulation of rRNA and mRNA expression.

**\*For correspondence:**
smitzlic@rz.hu-berlin.de

**Competing interest:** The authors declare that no competing interests exist.

## Editor's evaluation

This study brings a solid methodological advancement on previous attempts to understand the nature of the mitochondrial genome in *Toxoplasma*, applicable to Apicomplexa in general. The authors achieve extensive mitochondrial sequencing through a compelling and validated methodology, providing valuable RNA data. The improved catalog of mitochondrial rRNA and identification of overlapping protein-coding and rRNA genes will interest researchers in mitochondrial evolutionary biology and the mitoribosome community.

## Introduction

Apicomplexan parasites are a group of intracellular protozoan pathogens that can cause infectious diseases, including malaria and toxoplasmosis. Toxoplasmosis is caused by *Toxoplasma gondii*, one of the most widespread human parasites, with a seroprevalence of more than one-quarter of humans

worldwide (*Molan et al., 2019*). For many experimental questions, *T. gondii* remains an ideal model system for studying the molecular biology of Apicomplexa (*Szabo and Finney, 2017*). Given the economic and clinical impact of *T. gondii* and its apicomplexan relatives, there is continuous interest in distinctive biochemical and cellular features as potential targets for therapeutic intervention.

Apicomplexan cells possess a single mitochondrion that is respirationally active and essential for parasite survival (*MacRae et al., 2012*; *Melo et al., 2000*; *Seeber et al., 1998*; *Vercesi et al., 1998*). With only three exceptions, all mitochondrial proteins are nuclear-encoded and these nuclear genes contribute strongly to *P. berghei* and *T. gondii* cell fitness (*Bushell et al., 2017*; *Sidik et al., 2016*). Dozens of apicomplexan mitochondrial genomes have been sequenced (*Berná et al., 2021b*). These sequences showcase the extreme reductive evolution in apicomplexan mitochondria, setting records for the smallest mitochondrial genomes known to date, ranging in length from 6 to 11 kb (*Hikosaka et al., 2013*; *Oborník and Lukeš, 2015*). Mitochondrial genome organization is evolutionarily very variable, with some species having a linear, monomeric genome (*Babesia* spp.), while others have concatenated arrays of genomes (*Hikosaka et al., 2013*). The *T. gondii* mitochondrial genome is composed of 21 repetitive sequence blocks that are organized on multiple DNA molecules of varying lengths and non-random combinations (*Berná et al., 2021a*; *Namasivayam et al., 2021*). Apicomplexan mitogenomes code only for three proteins (COB, COX1, COXIII) and have highly fragmented genes for rRNAs (*Feagin et al., 2012*; *Seeber et al., 2020*). For example, in *P. falciparum*, 34 genes for rRNA fragments are scattered across the mitochondrial DNA on both strands of the genome without any linear representation of the full-length large or small subunit rRNA (*Feagin et al., 2012*). How the multitude of rRNA fragments are assembled into functional mitoribosomes in *T. gondii* remains unknown.

Since the electron transport chain in mitochondria is essential for the survival of apicomplexans, the expression of mitochondrial genes is likely to be essential as well. In fact, a nuclear-encoded RNA polymerase targeted to mitochondria has been shown to be essential in the blood stages of *Plasmodium spp.* (*Ke et al., 2012*; *Suplick et al., 1990*). Two mitochondrially localized RNA binding proteins from the RAP (RNA-binding domain abundant in apicomplexans) family are also essential for the survival of *P. falciparum*, although their role in RNA metabolism has not yet been determined (*Hollin et al., 2022*). In addition, the depletion of nuclear-encoded mitoribosomal proteins of *T. gondii* (*Lacombe et al., 2019*; *Shikha et al., 2022*) and *P. falciparum* (*Ke et al., 2018*) led to defects in the assembly of ETC complexes and in parasite proliferation, suggesting that mitochondrial translation is important for parasite survival. Resistance to the antimalarial drug atovaquone in *P. falciparum* and *T. gondii* has been linked to mutations in the *cob* (*cytB*) gene of mitochondria (*McFadden et al., 2000*; *Srivastava et al., 1999*; *Syafruddin et al., 1999*), further supporting the idea of active, essential translation in apicomplexan mitochondria.

Despite the clear importance of mitogenome expression in Apicomplexa, we lack a comprehensive catalog of transcripts and processing events. Genome-length, polycistronic transcripts are produced in *Plasmodium* that encompass all three open reading frames (*Ji et al., 1996*). The existence of full-length mRNAs was further confirmed by cDNA amplifications (*Namasivayam et al., 2021*) and long-read sequencing of mRNAs in both *Plasmodium* and *Toxoplasma* (*Lee et al., 2021*; *Namasivayam et al., 2021*). There is no evidence of dedicated transcription initiation for individual protein-coding genes, and it is assumed that mRNAs are processed from the polycistronic precursor (*Feagin et al., 2012*; *Hillebrand et al., 2018*; *Rehkopf et al., 2000*; *Suplick et al., 1990*), but the mechanism of processing and the required machinery are unknown. Similarly, the production of rRNA fragments is achieved post-transcriptionally by processing long precursor RNAs (*Ji et al., 1996*). A catalog of small RNAs, mostly rRNA fragments, was described for *P. falciparum* based on small RNA sequencing (*Hillebrand et al., 2018*), but in other apicomplexans, including *T. gondii*, our understanding of rRNA species is limited, and mostly based on predictions through sequence comparisons with *Plasmodium* (*Feagin et al., 2012*; *Namasivayam et al., 2021*).

We used a slightly modified protocol of mitochondria enrichment (*Esseiva et al., 2004*) to investigate the structure of the mitochondrial genome through long-read sequencing at an unprecedented depth. In parallel, we conducted high-throughput sequencing of small RNAs (<150 nt) and demonstrated that they are incorporated into polysomes. The combination of DNA sequencing results and transcriptome analysis also allowed us to identify previously undetected transcripts, many of which originate from block boundaries and represent fusions of coding and noncoding regions.

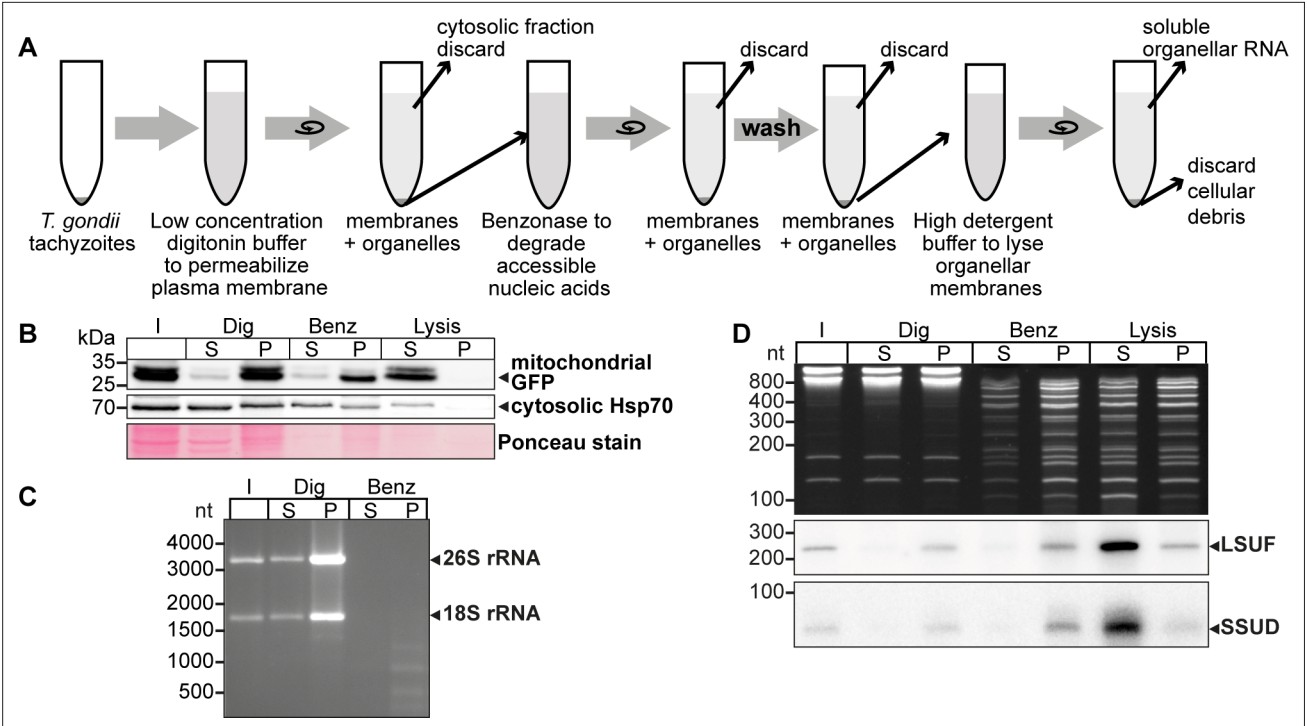

**Figure 1.** An experimental pipeline to enrich *T. gondii* mitochondrial RNAs. (**A**) *T. gondii* tachyzoites were harvested and incubated in a buffer with digitonin for plasma membrane permeabilization. Subsequently, accessible nucleic acids were digested by benzonase. After removal of the benzonase and washing the pellet, the intact organelles were lysed by a high detergent buffer. Soluble nucleic acids were separated by centrifugation and extracted from the supernatant. (**B**) *T. gondii* cells expressing a mitochondrial-targeted GFP were subjected to organelle enrichment. GFP was tracked by immunoblotting (5% volume of each fraction was analyzed). The GFP signal remained in the pellet and only shifted into the supernatant after the lysis step. In contrast, much of the cytosolic HSP70 is removed during the procedure, indicating specific enrichment of mitochondria. (**C**) RNA extracted from selected fractions of the organelle enrichment protocol was analyzed by agarose gel electrophoresis. (**D**) RNA extracted from selected fractions of the organelle enrichment protocol was analyzed on a denaturing 10% PAGE gel and blotted onto a nylon membrane. Radiolabeled DNA oligonucleotide probes were used to detect the mitochondrial rRNA fragments SSUD (from the small subunit of the ribosome) and LSUF (large subunit of the ribosome). Both fragments were found in pellet fractions after digitonin treatment, where they were protected from benzonase digestion. They only shifted to the supernatant after lysis. This demonstrates that the mitochondria stay intact during the procedure.

The online version of this article includes the following source data and figure supplement(s) for figure 1:

**Source data 1.** Raw gel and blot images.

**Figure supplement 1.** Small RNA (sRNA) sequence coverage of an 18 S rRNA gene.

## Results

### *T. gondii* organellar nucleic acids are enriched by a combination of selective membrane disruption and degradation of nucleo-cytosolic DNA and RNA

The *T. gondii* nuclear genome contains many insertions of mitochondrial DNA sequences (*Gjerde, 2013*; *Namasivayam et al., 2023*; *Ossorio et al., 1991*). To better distinguish between NUMTs (nuclear DNA sequences that originated from mitochondria) and true mitochondrial sequences, it is helpful to enrich mitochondrial DNA. We modified a cell fractionation method that takes advantage of the differential sterol content in plasma membranes and organellar membranes (*Esseiva et al., 2004*; *Subczynski et al., 2017*). We incubated cells with the detergent digitonin, which selectively permeabilizes sterol-rich membranes, leaving mitochondria and other organelles intact. We used a *T. gondii* strain that constitutively expresses GFP localized to the mitochondrial matrix to track the mitochondria during the purification process (*Figure 1A*). After permeabilizing the plasma membrane with digitonin, we treated the cells with benzonase to degrade nucleo-cytosolic nucleic acids. Following the inactivation of the benzonase, we lysed the mitochondria with high concentrations of detergents to release the soluble content.

We observed that mitochondrial GFP remained in the pellet fraction (P in *Figure 1B*) with only a minor signal in the supernatant, indicating that the mitochondria remained largely intact. In contrast, cytosolic HSP70 largely remained in the supernatant after digitonin and benzonase treatment, indicating that the method successfully reduced cytoplasmic proteins. Upon lysis of the mitochondria using Triton X-100 ('Lysis' in *Figure 1B*), the GFP signal was predominantly recovered in the supernatant, demonstrating efficient lysis of the mitochondria and release of soluble components. This fraction still contains HSP70, but slight contamination with cytosol is expected. We also found that cytosolic 18 S and 26 S rRNA were almost completely degraded, indicating the high efficacy of the benzonase treatment (*Figure 1C*). In summary, our enhanced protocol for organellar enrichment effectively depletes cytosolic protein and RNA components and enriches mitochondrial macromolecules.

To assess the integrity of mitochondrial RNA following our protocol, we utilized small RNA gel blot hybridization for two known mitochondrial rRNA fragments, SSUD and LSUF (*Namasivayam et al., 2021*). Our analysis of total nucleic acids in the gel revealed effective removal of major rRNA species by benzonase treatment (*Figure 1C*), while smaller degradation products accumulated (*Figure 1D* lanes 'Benz' and 'Lysis'). In contrast, we observed clear bands for both SSUD and LSUF, indicating their protection from benzonase degradation. Notably, we also detected signals for both rRNA fragments in the supernatant of the lysis fraction, confirming successful mitochondrial lysis and recovery of mitochondrial rRNA fragments.

## Long-read sequencing of DNA from enriched mitochondrial fractions identifies prevalent sequence block combinations

For the analysis of mitochondrial sRNAs, we needed a reference mitochondrial genome that would represent the existing sequence block combinations. A previous sequencing study used Oxford Nanopore sequencing technology (ONT) to identify combinations of sequence blocks in *T. gondii* mitochondria (*Namasivayam et al., 2021*). To improve read depth, we performed ONT sequencing on DNA gained from our improved organelle enrichment protocol. After filtering against the nuclear genome, we found 86,761 reads that had similarities to the published mitochondrial genome sequence (*Namasivayam et al., 2021*). This corresponds to 78.5 Mbp, which accounts for 9.2% of the total sequences that we attribute to the nucleus and mitochondria (*Supplementary file 1*). This is a 42-fold increase in the sequencing depth of the *T. gondii* mitochondrial genome compared to previous attempts (*Supplementary file 1*), which can be attributed to the effectiveness of the purification process. The length of mitochondrial reads ranged from 87 nt to 17,424 nt (*Supplementary file 2*). The GC content of these reads, on average 36.4%, further supports their mitochondrial origin (*Berná et al., 2021a*). We annotated the reads using published sequence block information (*Namasivayam et al., 2021*) and confirmed the previously described 21 sequence blocks, designated by letters from A-V (*Namasivayam et al., 2021*). We searched the sequencing data for repeated sequence elements but found no additional blocks, suggesting that the *T. gondii* mitochondrial genome is fully covered. We compared our reads with published ONT reads for the *T. gondii* mitochondrial genome (*Berná et al., 2021a*; *Namasivayam et al., 2021*). While smaller reads of our dataset are found in full within longer reads in the published datasets, we do not find any examples for reads that would be full matches between the datasets. This suggests that while the block content is identical between the different sequencing results, the actual block order was reshuffled by recombination, indicating the rapidity of mitogenome evolution in *T. gondii*, thus corroborating at a deeper sequencing coverage conclusions so far gained from a more limited read set (*Berná et al., 2021a*; *Namasivayam et al., 2021*).

The depth of our sequencing enabled us to quantify the genome block combinations. We first asked whether there is a bias in sequential block succession. We observed that the rate of occurrence of blocks varies dramatically in our dataset, with the most frequent block, J, occurring 45,849 times and the least frequent, C, occurring 3138 times (*Figure 2A*, *Supplementary file 3*). This suggests that differences in block combinations can be expected. After counting block combinations, we found that only a small fraction of all possible block combinations are prevalent within the genome (*Figure 2—figure supplement 1A*, *Supplementary file 4*), confirming block combinations observed previously (*Namasivayam et al., 2021*) and adding combination frequencies based on higher read numbers. For example, the most frequent block combination is J-B, which occurs 19,622 times in our reads. In total we identified 84 combinations of which 52 occur less than 50 times and make up less than 0.06% of

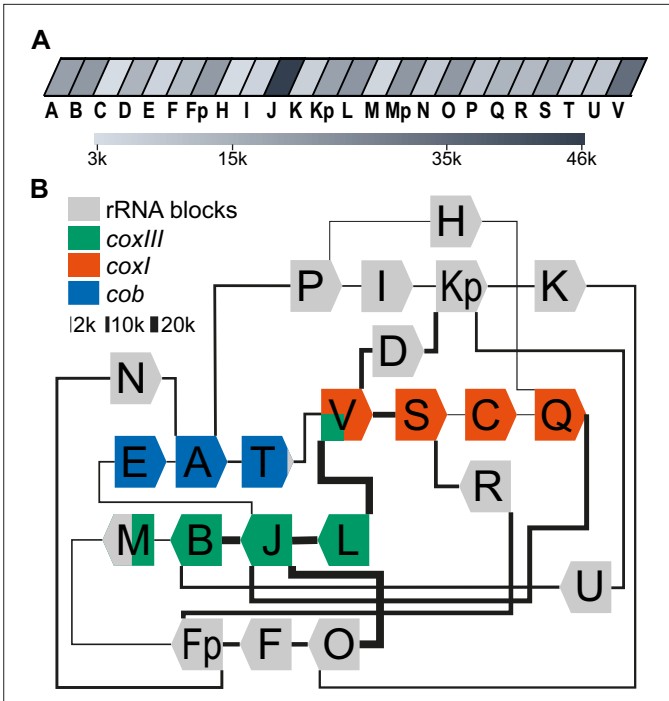

**Figure 2.** Long-read sequencing shows a specific system of block combinations. (**A**) Oxford Nanopore sequencing technology (ONT) DNA sequencing results were analyzed for the number of block occurrences. The gray scale indicates how many times each of the mitochondrial sequence blocks was found in the mitochondrial ONT reads. (**B**) Map of the *T. gondii* mitochondrial genome representing block combinations occurring in mitochondrial ONT reads. The thickness of the connecting lines indicates the absolute number of occurrences for each combination.

The online version of this article includes the following figure supplement(s) for figure 2:

**Figure supplement 1.** Counting of *T. gondii* mitochondrial sequence block organization.

**Figure supplement 2.** Proposed changes to the sequence blocks previously annotated by *Namasivayam et al., 2021* [1].,,,,,

the total number of combinations found (*Supplementary file 4*). Fourteen blocks (R, N, M, F, L, U, I, C, K, D, H, E, T) are always found with the same preceding and following block. The direction of the blocks in these combinations is fixed (for example, the 3'-end of the L block is always fused to the 5'-end of the J block, never to the 3'-end of the J-block). This indicates that the genome's flexibility is limited and that not all block combinations are realized (see also *Namasivayam et al., 2021*). Variability in genome structure is caused primarily by blocks S, O, Q, B, and P, which have up to three possible neighbors, and by blocks Fp, Kp, V, A, and J, which have more than three possible neighbors. All combinations are well covered in our ONT results and helped to refine block borders relative to previous annotations (*Figure 2—figure supplement 1*, *Figure 2—figure supplement 2*). Using the 32 high-frequency block combinations we found, we generated a map centered on the protein-coding genes (*Figure 2B*). We will later show that all blocks not encoding proteins express rRNA fragments and are thus called rRNA blocks here. The map is not designed to represent physically existing DNA molecules, since the sequencing technology used cannot detect branched or circular DNAs, nor are the blocks drawn to scale. However, it helps to indicate the complexity of the nonrandom recombination events shaping the genome. For example, when considering the positioning of the three protein-coding genes within this map, it is conspicuous that their three 5'-ends are always flanked by blocks also encoding a protein (*Figure 2B*), although it is unclear whether this is of functional relevance.

Full-length coding regions had been found previously on nanopore reads in *T. gondii* (*Namasivayam et al., 2021*). In our improved representation of the mitogenome, we identified a large number of full representations of *cob*, *coxI*, and *coxIII* in our dataset (*cob*: 1612; *coxI*: 1404; *coxIII*: 1487). Thus, of all reads long enough to carry one of the following ORFs, 5.1% contain full-length coxIII, 8.6% contain full-length cob, and 11.1% full-length cox1, respectively (*Supplementary file 5*). This may

be adequate for the expression of the encoded proteins; however, we cannot currently exclude the possibility of genomic or post-transcriptional block shuffling that could lead to more complete open reading frames (as discussed in *Berná et al., 2021b*). We next applied approaches established previously to represent biased recombination events based on alternative block combinations (see Figure S12 in *Namasivayam et al., 2021*) to our improved ONT read set. For example, block S, as part of the *coxI* coding region, was followed in 80% of its occurrences by the rRNA block R and only in 20% by the next coding block C (*Figure 2B*, *Figure 2—figure supplement 1B*). Similarly, block B, as part of the *coxIII* reading frame, was followed in 66% of all cases by the rRNA block U and less frequently by the terminal block of *coxIII*, which is block M. Whether such fusions of coding and non-coding block is of functional relevance, was unclear so far.

## Small RNA sequencing identifies a comprehensive set of mitochondrial rRNA fragments

For *T. gondii* mitochondria, rRNA species were previously predicted based on sequence comparisons to *Plasmodium* rRNA sequences (*Namasivayam et al., 2021*). We used Illumina-based small RNA sequencing of *T. gondii* to identify mitochondrial rRNA fragments. We sequenced both the mitochondria-enriched fraction and the input fractions described in *Figure 1*. In the enriched fraction, we mostly retrieved reads for cytosolic rRNA degradation products and poor coverage of mitochondrial sequences (not shown). This is likely caused by the benzonase step in the protocol, which generates an abundance of small degradation products of cytosolic rRNA (*Figure 1D*; *Figure 1—figure supplement 1*). In the input samples, intact full-length rRNAs are removed by selecting small

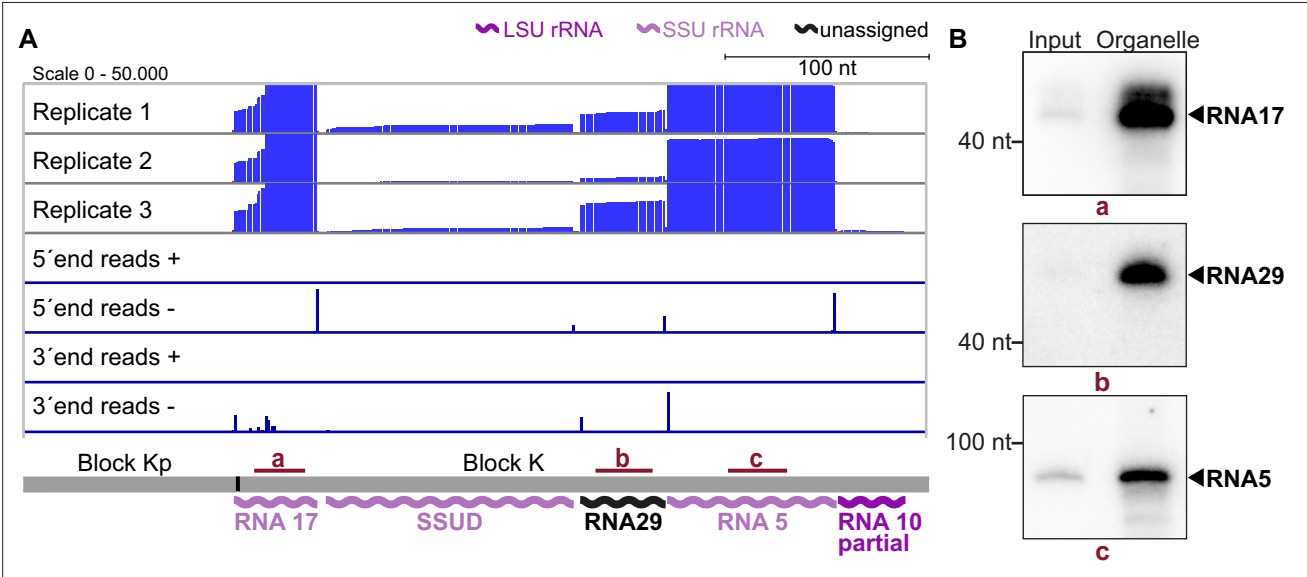

**Figure 3.** Small RNA sequencing identifies mitochondrial small RNAs. (**A**) Upper three rows: excerpt of mapping results in three strand-specific small RNA sequencing replicates starting from total input RNA from our *T. gondii* organelle RNA preparation. Read depth was counted by the number of reads at each position. Note that the 5'-ends of the RNAs shown here are on the right. Lower four rows: only the terminal nucleotides of each read on the plus and minus strands of the genome are shown. The size and position of genes below the coverage graphs are all drawn to scale. Red bars with lowercase letters indicate the positions of probes used for detecting RNAs in RNA gel blot hybridizations shown in (**B**). (**B**) Equal quantities of total input RNA from *T. gondii* and from organelle preparations were loaded onto a denaturing PAGE gel and analyzed by RNA gel blot hybridization with the probes indicated.

The online version of this article includes the following source data and figure supplement(s) for figure 3:

**Source data 1.** Raw blot images.

**Figure supplement 1.** Detection of LSUD/E and LSUF/G transcripts.

**Figure supplement 1—source data 1.** Raw blot images.

**Figure supplement 2.** Overview of non-coding RNAs identified in *T. gondii* mitochondria.

**Figure supplement 2—source data 1.** Raw blot images.

**Figure supplement 3.** Mapping of small RNAs onto exemplary Oxford Nanopore sequencing technology (ONT) sequencing reads.

RNAs during library preparation, resulting in less prevalent cytosolic rRNA reads. We bioinformatically removed the remaining cytosolic rRNA reads as well as any reads shorter than 20 nt. Poly-A stretches at the 3'-ends were clipped as well. Finally, we mapped the remaining reads against the mitochondrial genome (*Supplementary file 6*) using the block combinations identified here by ONT sequencing (*Figure 2*).

The results of our mapping study show that the 5'-termini of mitochondrial small RNAs are clearly defined, making it easy to identify their transcript ends (as shown in *Figure 3A*). In agreement with findings from previous studies on small RNAs from *P. falciparum* (*Hillebrand et al., 2018*), we found that the 3'-ends of many transcripts were more variable (as seen in the example of RNA17 in *Figure 3A*). We identified a total of 34 small RNAs that accumulate in *T. gondii* mitochondria (*Supplementary file 7*). Among these, 11 correspond to previously predicted rRNA fragments (*Namasivayam et al., 2021*). Our sequencing data confirmed that these sequences are expressed and allowed us to refine the exact rRNA fragment ends (*Supplementary file 7*). Additionally, our small RNA sequencing revealed some larger differences compared to previous transcript predictions. This included a reassignment of SSUF to the opposite strand and also affected the four rRNA fragments LSUF, LSUG, LSUD, and LSUE, which had been predicted as separate transcripts of the large subunit (LSU) of the ribosome (*Namasivayam et al., 2021*). Our sequencing results suggest that there is an accumulation of transcripts containing LSUF and LSUG regions and LSUD and LSUE regions, respectively. Both of these transcripts were verified via northern blotting (*Figure 3—figure supplement 1*). The longer transcripts were found to be much more abundant than smaller transcripts that were also detected (*Figure 3—figure supplement 1*), suggesting that the longer transcripts represent the functional rRNA fragments in *T. gondii* mitochondria. Knowing the exact sequence of rRNA fragments is crucial for further investigations into the structure of mitoribosomes, as it is key for predictions of secondary structures.

## Discovery of previously undetected rRNA fragments

### sRNAs cover most of the noncoding sequence blocks of the *T. gondii* genome

Out of the 34 small RNA fragments identified, 23 have not been described previously for *T. gondii* (see *Supplementary file 7* and *Figure 3—figure supplement 2*). 17 of the 23 are homologs of sRNA fragments in the apicomplexans *P. falciparum* and *E. leuckarti* (marked in bold in *Supplementary file 7*). These were named according to their *Plasmodium* homologs (*Feagin et al., 2012*). Six further sRNAs are exclusively conserved within cyst-forming Eucoccidians, the closest apicomplexan relatives of *T. gondii* (last column in *Supplementary file 7*). We assigned numbers to these sRNAs extending the *Plasmodium* nomenclature (*Feagin et al., 2012*). We next asked, how many of all 34 sRNAs have homologies to rRNA from *E. coli*. We could find twelve LSU homologs and eleven SSU homologs in accordance with previous analyses in *P. falciparum* (*Feagin et al., 2012*; *Hillebrand et al., 2018*; *Supplementary file 8*). Only for the *P. falciparum* LSU fragment LSUC and the SSU fragment RNA12, we were unable to identify corresponding homologs in *T. gondii*.

We next analyzed the accumulation of selected RNAs previously undetected in *T. gondii* for the sequence blocks Kp-K: Transcripts RNA5, 17, and 29, as well as the already predicted transcripts RNA10 and SSUD are found on the minus strand strandedness according to *Namasivayam et al., 2021*; MN077088.1 - MN077111.1. They are separated by single nucleotides in the mitochondrial genome. In this and most other noncoding blocks, the DNA sequence is almost fully used for transcript production (see *Figure 3A*, *Figure 3—figure supplement 2*, *Figure 3—figure supplement 3*). To validate the sequencing data, we performed RNA gel blot hybridization using probes against RNA17, RNA29, and RNA5 (*Figure 3B*) and indeed detected transcripts of the expected size (41 nt, 42 nt, and 83 nt, respectively). The transcripts were more abundant in our organelle preparations than in input samples, which indicates that they are of organellar origin and did not originate from NUMTs. The sequencing and RNA gel blot efforts demonstrate that almost the entire sequence of the blocks Kp-K is represented by small RNAs.

### Dual use of block borders for sRNA production

Among the small RNAs identified here, there is also a class that was only detectable due to our insights into genome block combinations. Using block combinations for mapping analysis, we identified 15

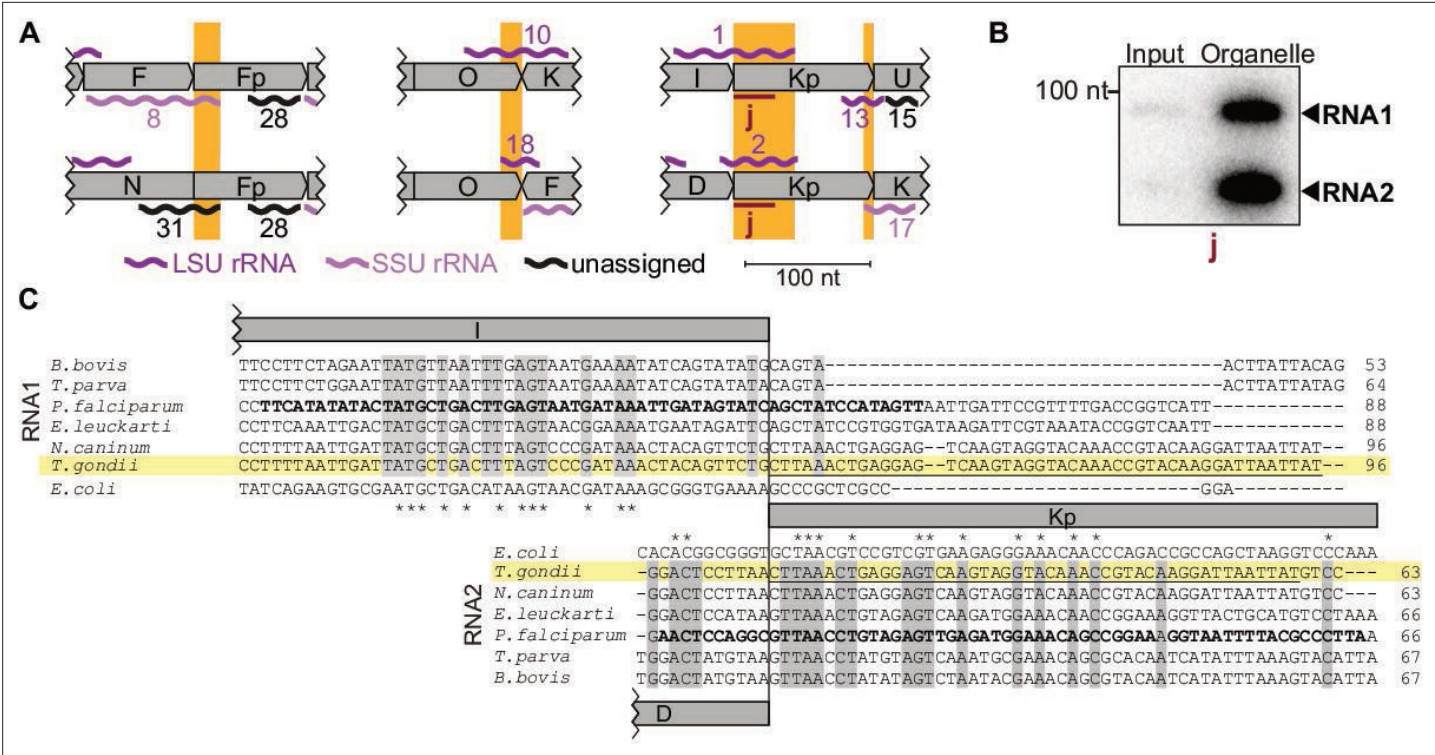

**Figure 4.** Noncoding RNAs at block borders. (**A**) Schematic representation of three block borders that are used by different transcripts (sequences used for two transcripts are marked by orange rectangles). Transcripts are shown as wiggle lines and are named according to prior nomenclature (**Namasivayam et al., 2021**). Assignment of transcripts to the large and the small rRNA are indicated by color. The red bars with the lowercase letter 'j' indicate the position of the probe used for detecting RNA1 and RNA2 by RNA gel blot hybridization. LSU/SSU = large/small subunit of the ribosome. (**B**) RNA gel blot hybridization of RNA1 and RNA2 using RNA extracts from input and organelle-enriched fractions of the organelle enrichment protocol is shown in **Figure 1**. Equal RNA amounts were loaded. (**C**) Alignment of RNA1 (top) and RNA2 (bottom) sequences from different apicomplexan species. Sequence excerpts from *E. coli* 23 S rRNA (1240–1301 for RNA1, 944–1010 for RNA2) corresponding to sections of the alignment were added manually based on previous analyses (**Feagin et al., 2012**). Nucleotide positions conserved in all apicomplexans are shaded in gray. The residues in the *P. falciparum* sequence suggested to replace sequence elements in the *E. coli* 23 S rRNA structure (**Feagin et al., 2012**) are marked in bold. Residues conserved between apicomplexans and *E. coli* are marked with asterisks. Note that RNA1 and RNA2 are aligned according to their shared sequence in genomic block Kp.

The online version of this article includes the following source data for figure 4:

**Source data 1.** Raw blot images.

transcripts that span two blocks, i.e., there are sRNAs at almost half of all 31 block borders. Often, we find sRNAs that share a block border sequence at one end, but differ at the other end depending on the block combination (see **Figure 4A**, orange boxes). An example of sequences shared at the 5'-end is represented by the pair RNA8 and RNA31, which both start in block Fp but end in different blocks (**Figure 4A**). An example of block sharing at the 3'-end is the RNA pair RNA1/RNA2. Both RNAs terminate at very similar positions in block Kp but start in different blocks (**Figure 4A**). Using a probe for the common 3'-part in block Kp, we detected both RNA1 and RNA2 in an RNA gel blot hybridization experiment (**Figure 4A and B**). Both RNA1 and RNA2 have homology to *E. coli* LSU in their 3'-portion via block Kp, but only RNA1 has additional homology to *E. coli* LSU via block I (**Figure 4C**; **Feagin et al., 2012**). It is interesting to consider that RNA1 could, therefore, be used in two positions in the large subunit of the ribosome. In conclusion, block combinations can lead to the expression of RNAs in *T. gondii* that are not found in apicomplexan species with a simpler genome organization (**Supplementary file 7**).

## Coding regions contribute to sRNAs at block borders

In addition to the three pairs of overlapping, noncoding RNAs described above (**Figure 4A**), we also found five RNAs that combine sequences from coding and non-protein-coding blocks (**Figure 5A**

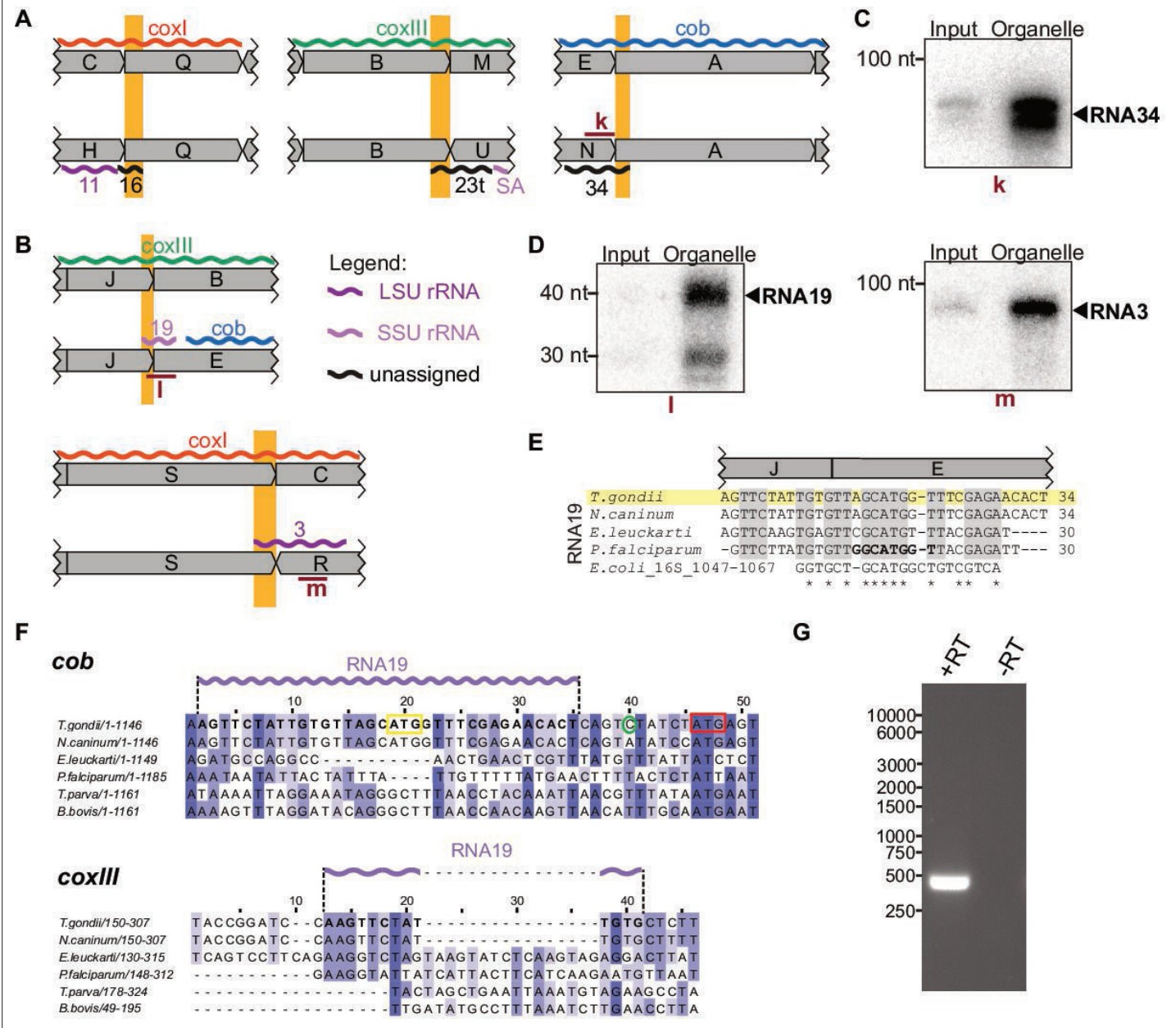

**Figure 5.** Small RNAs that consist of coding and non-protein-coding sequences at block borders. (**A**) Schematic representation of the genomic position of three RNAs that are partially antisense to coding sequences. (**B**) Schematic representation of the genomic position of two RNAs that partially overlap with coding sequences. (**C**) RNA gel blot hybridization of RNA34. Equal amounts of RNA extracts from input and organelle-enriched fractions were analyzed. (**D**) RNA gel blot hybridization of RNA19 and RNA3 – for details see (**E**) Alignment of *T. gondii* RNA19 with homologous sequences from other apicomplexan species and from *E. coli*. (**F**) Alignment of a section of the *coxIII* and *cob* sequences from different apicomplexan species. Alignments were prepared using Clustal Omega (https://www.ebi.ac.uk/Tools/msa/clustalo/). Nucleotides of *T. gondii* RNA19 are marked in bold in both alignments and as a lilac wiggle line above the sequences. The sequences are interrupted by indels in *coxIII* and *cob* of other apicomplexans that have stand-alone RNA19 sequences (e.g. *P. falciparum*). Within the *cob* sequence, the RNA19 sequence covers an annotated start codon (*Alday et al., 2017*, yellow box; *Namasivayam et al., 2021*). However, an alternative start codon downstream has been suggested as well (red box, *McFadden et al., 2000*) and is in fact where sequence conservation starts. The green circle indicates the *cob* 5′ transcript terminus as determined by rapid amplification of cDNA ends (RACE) (see G). (**G**) Agarose gel analysis of 5′ RACE PCR products was performed to identify the *cob* 5′ transcript end. +RT and -RT indicates the presence or absence of reverse transcriptase, respectively.

The online version of this article includes the following source data and figure supplement(s) for figure 5:

**Source data 1.** Raw gel and blot images.

**Figure supplement 1.** Alignment of RNA3 sequences from different apicomplexan species.

*and B*). Among them, RNA16, RNA23t, and RNA34 were partially antisense to mRNAs (*Figure 5A*), and we confirmed the accumulation of RNA34 by RNA gel blot hybridization (*Figure 5C*). None of the three RNAs had detectable homologies to *E. coli* rRNA based on simple sequence searches, but structural conservation cannot be ruled out. Mitochondrial rRNA sequences from kinetoplastids and diplonemids show very little sequence conservation but are still part of the mitoribosome (*Valach et al., 2023*; *Ramrath et al., 2018*), suggesting that future analyses might uncover hidden rRNA similarities. With the exception of RNA34, the RNAs representing a fusion of non-protein-coding regions and protein-coding regions have homologous sequences in the mitochondrial genome of *P. falciparum* (*Feagin et al., 2012*). In *P. falciparum*, however, none of them are antisense to coding regions. Whether these sRNAs form sRNA:mRNA interactions and whether this has functional consequences remains to be investigated.

Two of the small RNAs contained coding sequences in the sense direction (*Figure 5B*). Both were validated by RNA gel blot hybridization (*Figure 5D*). RNA19 contains the *coxIII* coding sequence adjacent to the *cob* coding region by a combination of block J and block E (*Figure 5E*). *Cob* has two in-frame start codons (*Figure 5F*). Using rapid amplification of cDNA ends (RACE), we showed that the 5'-end of the *cob* mRNA is 6 nt upstream of the second start codon and 10 nt downstream of RNA19 (*Figure 5F and G*). Being only four nucleotides apart, the 3'-end of RNA19 is possibly generated during the formation of the 5'-end of *cob*. RNA19 is conserved in other apicomplexan mitochondrial genomes (*Figure 5E*) but is not part of open reading frames in *Plasmodium*. Homologies and a structural fit to the SSU rRNA of *E. coli* have been noted for *P. falciparum* RNA19 (*Figure 5E*, *Feagin et al., 2012*). This sequence similarity to rRNA is maintained in *T. gondii* (*Figure 5E*), which suggests that despite overlapping with *coxIII* coding sequence at the J-E block border, RNA19 is functional. It is remarkable that this sequence serves dual purposes for coding a protein and an rRNA. There is only low sequence conservation of the *coxIII* sequence used by RNA19 (*Figure 5F*), which might have allowed the evolutionary acquisition of an additional use as an rRNA fragment. In addition to RNA19, a second RNA-encompassing coding sequence is RNA3, which is situated at the S-R block border (*Figure 5B*). Block S is part of the *coxI* coding region - its terminal 22 nucleotides contribute to RNA3. The similarity to *E. coli* 23 S rRNA is, however, restricted to the noncoding block R (*Figure 5—figure supplement 1*). The Block S sequence in RNA3 is, therefore, a 5'-extension of this rRNA, and it remains to be determined whether this is of functional consequence for the mitochondrial ribosome. It is noteworthy that the 5' end of RNA3 is located within a region of the 23 S rRNA secondary structure that exhibits low conservation (*Feagin et al., 2012*), which may suggest that its overlap with coding regions is potentially tolerable.

In sum, these results suggest that at block borders, *T. gondii* makes dual use of several protein-coding sequence blocks. Depending on the respective block combination, they are either part of the protein-coding sequence or can code for an rRNA fragment.

## Mitochondrial small RNAs are part of a large-molecular weight complex

The apicomplexan mitochondrial ribosome with its peculiar organization has not been characterized in any detail. There is strong evidence from genetic and pharmacological studies that translation occurs in the mitochondria of apicomplexans (*Alday et al., 2017*; *Lane et al., 2018*; *McFadden et al., 2000*; *Vaidya et al., 1993*). Additionally, the presence of a large Megadalton complex containing nuclear-encoded ribosomal proteins targeted to mitochondria has been shown by blue native gel electrophoresis (*Lacombe et al., 2019*). However, a comprehensive catalog of mitoribosomal constituents is missing, and the extent to which rRNA fragments are part of mitoribosomes remains unanswered, nor have polysomes been identified as a direct readout for active ribosomes. To ascertain the presence of sRNAs within high-molecular weight complexes, we fractionated enriched organellar preparations using sucrose gradient analysis. We employed conditions known to preserve mitochondrial ribosomes (*Waltz et al., 2021a*) conditions established to dissociate ribosomes (10 mM EDTA, no MgCl$_2$, 300 mM KCl). Should sRNAs be integral to ribosomes, their dissociation would be expected to result in a migration towards lower molecular weight fractions.

RNA extracted from the gradient fractions was initially analyzed using Ethidium bromide-staining in denaturing polyacrylamide urea gels (*Figure 6A*). The paucity of detectable signals aligns with the expected degradation of abundant cytosolic rRNAs during preparation with benzonase. Notably, only the top fractions of the gradient (fractions 1–6) displayed bands, while the deeper fractions showed

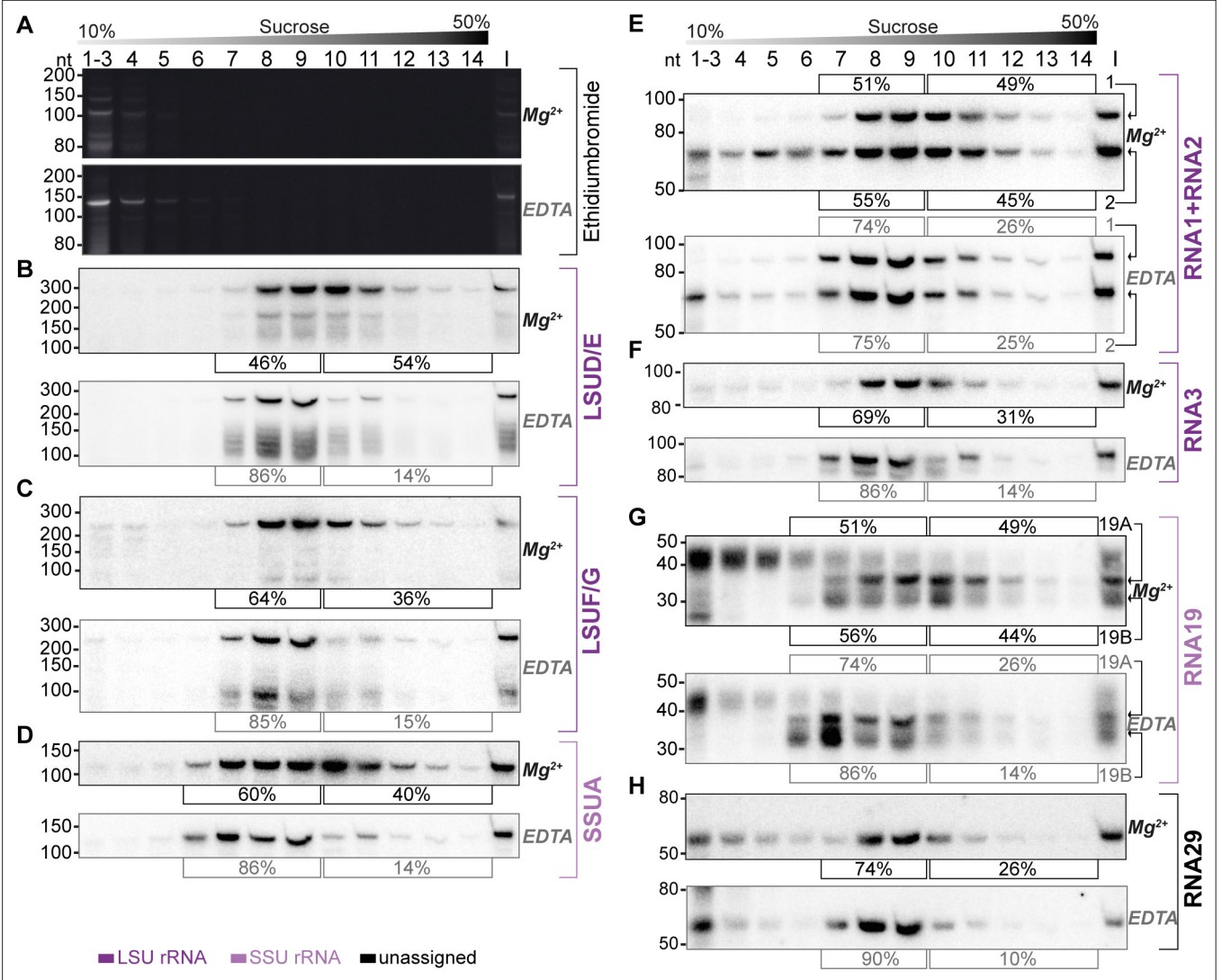

**Figure 6.** Sucrose density centrifugation analysis demonstrates the association of mitochondrial small RNAs with large molecular weight complexes. *T. gondii* organelle-enriched extracts were fractionated by sucrose density gradient centrifugation and subsequently analyzed by RNA gel blot hybridization. Buffer conditions during the experiment were either 30 mM $MgCl_2$, 100 mM KCl (labeled $Mg^{2+}$ next to the blots), or 10 mM EDTA, 300 mM KCl (labeled EDTA next to the blots). (**A**) Prior to blotting, the gels were stained with ethidium bromide. Atop the gel images, a schematic representation illustrates the sucrose density gradient. Subsequent panels display a series of RNA gel blot hybridization assays. These assays were conducted on blots derived from the two ethidium bromide-stained gels presented in (**A**). Following each probe, the blot was stripped of its signal by employing a denaturing buffer. The oligonucleotides in these assays targeted LSUD/E (**B**), LSUF/G (**C**), SSUA (**D**), RNA1 +RNA2 (**E**), RNA3 (**F**), RNA19 (**G**), and RNA29 (**H**). Percentages in square brackets encircling each blot indicate the proportion of signal detected in fractions 7–9 or 6–9 for large subunit (LSU) or small subunit (SSU) small RNAs (sRNAs), respectively, versus signal intensity in fractions 10–14. 'I' corresponds to 10% of the lysate prior to gradient loading. The RNA names are color-coded to indicate homology: deep purple for RNAs with homology to *E. coli* rRNA from the large ribosomal subunit, light purple for those from the small subunit, and black for RNAs without previously noted homology.

The online version of this article includes the following source data and figure supplement(s) for figure 6:

**Source data 1.** Raw blot images.

**Source data 2.** Raw blot images.

**Figure supplement 1.** Migration patterns of mitochondrial small RNAs (sRNAs) in sucrose density gradients under $Mg^{2+}$-depleted conditions.

no discernible signals, reaffirming the effectiveness of the organellar preparation in eliminating cytosolic contaminants. The observed bands in the top fractions may represent degradation products of prevalent rRNA species, released from ribosomes by benzonase but persisting in the organellar preparation due to their abundance. Subsequent to gel electrophoresis, we transferred the RNA onto

membranes and hybridized them with probes targeting seven specific sRNA species. LSUD/E and LSUF/G, along with RNA1, RNA2, and RNA3, were chosen for analysis due to their sequence similarities with *E. coli* 23 S rRNA, as noted by *Feagin et al., 2012*. Similarly, SSUA and RNA19 were chosen since they resemble parts of *E. coli* 16 S rRNA (*Feagin et al., 2012*). Additionally, RNA3 and RNA19 were tested as they utilized the coding sequences of coxI and coxIII, respectively. Finally, we also included RNA29, an sRNA not yet assigned to the ribosome based on sequence similarity.

We observed the peak signals for all sRNAs in the middle of the gradient, but specifically not in fractions 1–4, where free sRNAs would be expected to localize, akin to the signals observed in the ethidium bromide-stained samples (*Figure 6*). Second, all tested sRNAs were found to also migrate into the deepest gradient fractions 12–14. Collectively, these observations strongly indicate that the sRNAs in question are predominantly part of high molecular weight complexes. To evaluate the influence of ribosome-dissociative conditions on the fractionation of sRNAs in sucrose gradients, we quantified the sRNA signals in the deeper fractions (10–14) compared to the peak fractions (7–9 for LSU rRNAs and 6–9 for SSU rRNAs). For all sRNAs examined, a shift was observed from the deeper to the earlier fractions under dissociative conditions (*Figure 6*). This shift indicates that the large molecular weight complexes, with which these sRNAs are associated, are susceptible to ribosome-dissociative conditions. The most parsimonious conclusion is that all sRNAs tested are rRNAs.

We then asked whether rRNA fragments with homology to *E. coli* 23 S versus those with homology to 16 S rRNA exhibit differential signal distribution. We found that rRNAs presumed to be small subunit rRNAs, namely SSUA and RNA 19, displayed signals in fraction 6 (*Figure 6D and G*), a pattern not observed or less pronounced in presumed LSU rRNAs such as LSUD/E, LSUF/G, RNA1 +2, and RNA3 (*Figure 6B, C, E and F*). The SSU-specific signal in fraction 6 persisted under both standard and dissociative conditions, implying that it represents free SSU (*Figure 6D and G*). We quantified the signals in all fractions under dissociative conditions and compared the profiles of sRNA signals across the gradient (*Figure 6—figure supplement 1*). This shows that under dissociative conditions, the distribution and peaks differ markedly between the four predicted LSU rRNAs and the two SSU sRNAs (*Figure 6—figure supplement 1*). The distinct distribution patterns for rRNAs with homology to LSU versus SSU rRNA suggest the gradient allows to assign individual rRNA fragments to the two subunits of the ribosome. Indeed, the previously unclassified RNA29 exhibited a distribution pattern very similar to LSU rRNAs (*Figure 6H*). It showed significantly less signal in fraction 6 than SSUA and a distribution across the gradient similar to LSUF/G and LSUD/E (*Figure 6C and D* and *Figure 6—figure supplement 1*). This pattern indicates that RNA29 likely belongs to the LSU category. Unlike most other rRNAs, RNA29 also showed strong signals in the low molecular weight fractions 1–5 (*Figure 6H*), hinting at a proportion of unassembled RNA29 in our preparation. Such un-assembled rRNA was also observed for RNA2, while the cognate RNA1 was barely detected in lower molecular weight fractions (*Figure 6E*). It remains to be determined whether different assembly kinetics or degradation rates contribute to the RNA signals in the upper parts of the gradient. Both RNA1 and RNA2 demonstrated deep penetration into the gradient, and showed a gradient distribution consistent with the LSU distribution pattern. Both RNA1 and RNA2 are sensitive to EDTA treatment. Together this leads to the conclusion that RNA1 and RNA2 are both assembled into ribosomes.

We furthermore observed that despite containing sequences derived from mRNAs, both RNA19 and RNA3 are incorporated into ribosomes (*Figure 6B*). For RNA19, several transcripts were found. The largest isoform is mainly found in low-molecular weight fractions. Conversely, the smaller RNA19 transcripts are observed to migrate into the high-molecular weight fractions, which is EDTA-sensitive (*Figure 6G*). Our findings suggest that RNA19 and RNA3 are incorporated into the ribosome.

## Mitochondrial mRNAs show an association with high-molecular weight complexes that are sensitive to ribosome-dissociative conditions

We next analyzed the distribution of mitochondrial mRNAs in a sucrose gradient, under non-dissociative and dissociative conditions. The extended length of these mRNAs facilitates their detection using qRT-PCR, a method that is impractical for the analysis of many shorter sRNAs. To ensure the gradient exhibited characteristics akin to those used in the sRNA distribution analysis (*Figure 6*), we concurrently analyzed the sRNA LSUF/G. This confirmed that LSUF/G migrates deep into the gradient, which is sensitive to ribosome-dissociative conditions (see *Figures 6C and 7A*).

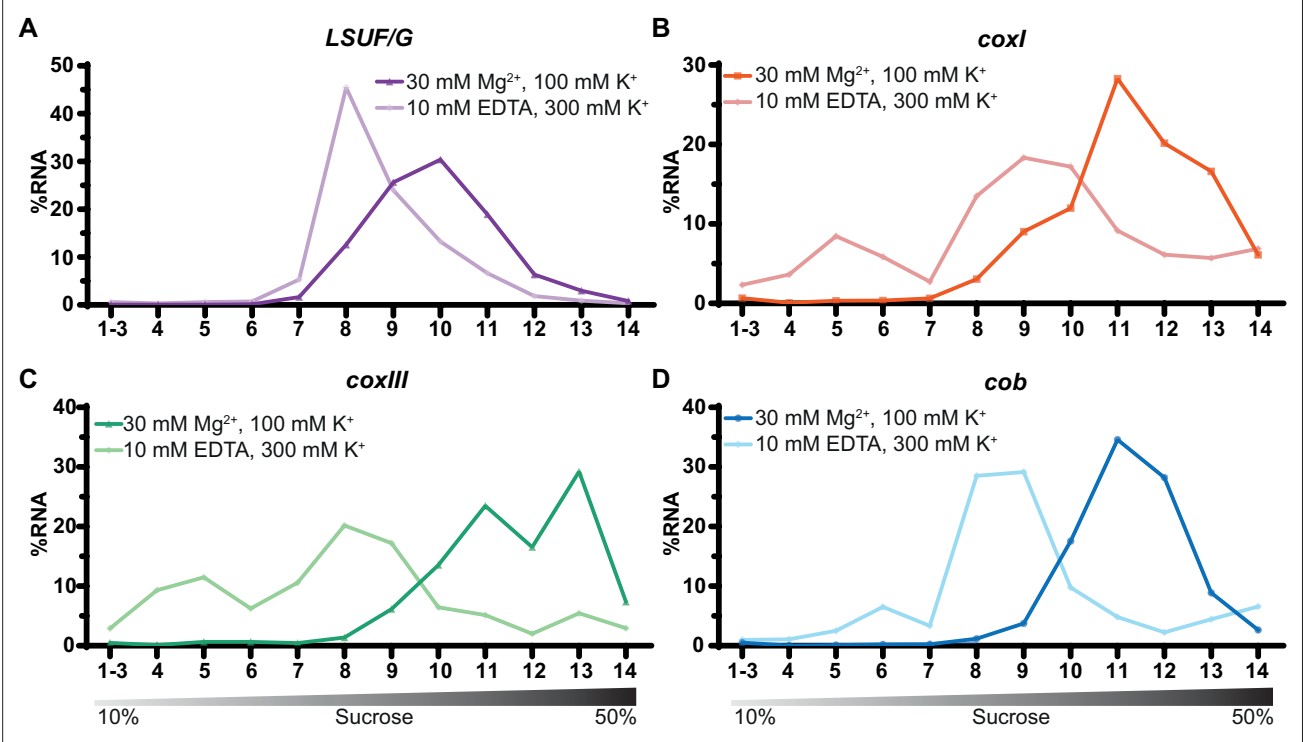

**Figure 7.** Sucrose density centrifugation analysis of mitochondrial mRNAs. *T. gondii* organelle-enriched extracts treated with standard salt concentrations or with an Mg$^{2+}$-depleted buffer, respectively, were fractionated by sucrose density gradient centrifugation and subsequently analyzed by RT-qPCR. Data was normalized using human estrogen receptor 1 (ESR1) as an external spike-in control. Relative quantities per amplicon are visualized as percentage of RNA per fraction. All three mRNAs (shown in panel B-D) and one large subunit (LSU) rRNA fragment serving as a control (LSUF/G shown in panel A) were analyzed from a single sucrose gradient fractionation. The gradient density is shown schematically below the graphs.

All three mRNAs migrated deep into the gradient, reaching their peak in fractions 11–13, as depicted in *Figure 7B–D*. Upon treatment with EDTA, the mRNAs shifted towards fractions associated with lower molecular weights. We conclude that mitochondrial mRNAs in *T. gondii* are populated with ribosomes, which directly supports previous suggestions of active translation in apicomplexan mitochondria.

## The nuclear-encoded mitoribosomal protein L11 is found in EDTA-sensitive large complexes

We next sought to analyze how a ribosomal protein is distributed in our sucrose gradient with the goal to have an additional read-out for ribosomes and polysomes and to investigate whether this mirrors the distribution of sRNAs. We chose TGRH88_003980_t1, which was in previous in silico screens identified as a potential mitoribosomal protein L11 (*Gupta et al., 2014*; *Lacombe et al., 2019*). We used CRISPR/Cas9 to introduce an HA-tag to the C-terminus of *Tg*uL11m by homologous recombination (*Figure 8—figure supplement 1A and B*). The tagged protein is immunologically detected as a single band, demonstrating the specificity of the tagging experiment (*Figure 8—figure supplement 1C*). We next performed an immunofluorescence assay (IFA) to check the localization *Tg*uL11m:HA. The HA-signal co-localized with the mitochondrial marker protein *Tg*Tom40 (*van Dooren et al., 2016*) confirming the localization of *Tg*uL11m to *T. gondii* mitochondria (*Figure 8A*). We next analyzed the distribution of *Tg*uL11m:HA in sucrose gradients and found a main peak in fractions 9–11 with signals reaching into the deepest fractions of the gradients (*Figure 8B*). This mirrors the distribution of sRNAs from the large subunits like LSUD/E (*Figure 6B*). Upon EDTA treatment, the signal shifts from the deepest fractions (12-14) into lower fractions (8-10), suggesting that we observe polysomes in fractions 11–14 and that the large subunit of the ribosome is found in fractions 8–10.

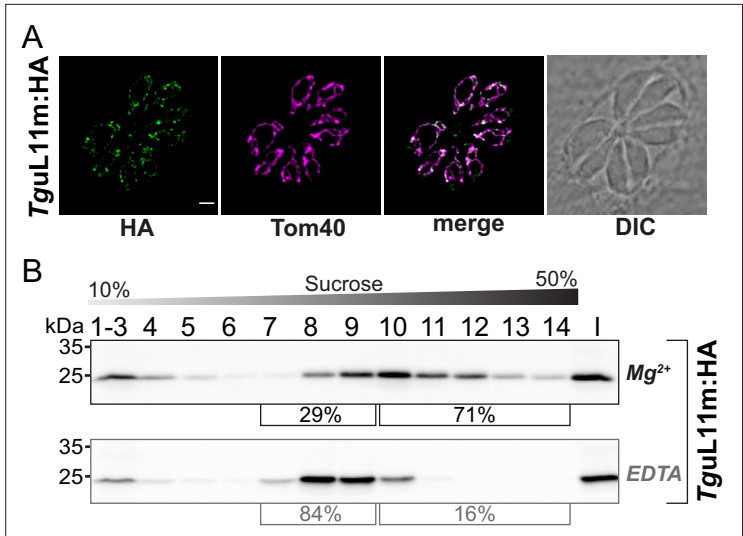

**Figure 8.** Analysis of the putative mitoribosomal protein L11. (**A**) *T. gondii* parasites expressing *Tg*uL11m:HA were subjected to immunofluorescence assays. These assays were conducted using an anti-HA antibody, visualized in the green channel. The mitochondria within these samples were identified using an antibody against the outer membrane protein Tom40, shown in the magenta channel. (**B**) For the analysis of HA-tagged mitochondrial ribosomal protein L11 (*Tg*uL11m:HA), sucrose density centrifugation was employed. Organelle-enriched extracts from *T. gondii* lines were either untreated (Mg$^{2+}$) or treated with EDTA. Subsequently, these extracts were fractionated via a sucrose gradient. The fractions obtained were analyzed using SDS-PAGE, followed by immunoblot analysis. *Tg*uL11m:HA was detected using an antibody against HA. Percentages in square brackets show the proportion of signal detected in fractions 7–9 versus signal intensity in fractions 10–14. 'I' indicates 10% of the input material for the centrifugation.

The online version of this article includes the following source data and figure supplement(s) for figure 8:

**Source data 1.** Raw blot images.

**Figure supplement 1.** Generation of a tagged ribosomal protein TgRH-uL11mHA.

**Figure supplement 1—source data 1.** Raw gel and blot images.

## Discussion

### Constraints in the recombination-active mitochondrial DNA of *Toxoplasma gondii*

The complexity of the *T. gondii* mitochondrial genome is puzzling, with an extensive variety of sequence block combinations and repetitions. The individual ONT reads are not repeated in the two previously published datasets (discussed in *Berná et al., 2021b*) and we also did not find an overlap between the data presented here and published previously (*Namasivayam et al., 2021*). Thus, the large number of block combinations identified here reinforces and elevates previous conclusions that continuous recombination shuffles the blocks (*Namasivayam et al., 2021*). At first sight, this might be considered a sign of evolutionary tinkering, similar to the mitochondrial idiosyncrasies found in other taxa that have been suggested to be non-adaptive, such as the fragmented mitochondrial genomes of diplonemids (*Burger and Valach, 2018*) or the repetitive and heterogeneous genomes of plant mitochondria (*Kozik et al., 2019*). Upon closer scrutiny, however, the reshuffling appears limited to specific block borders and is not random. In fact, the improved sequencing depth presented in this study revealed strong constraints on the allowed block combinations in *T. gondii* mtDNA. Thus, a limited number of allowed recombination sites generates the variety of actual DNA fragments sequenced. The question remains whether the constraints outlined in the comprehensive model of mitochondrial genome architecture presented here indicate functional features.

### The purpose of genome blocks in light of small RNA production

Our small RNA sequencing results revealed that 15 small RNAs span block borders. There are different explanations for this peculiar localization of small RNAs. One possibility is that these RNAs are involved

in the DNA recombination and replication process that occurs at block borders. RNA molecules have been found to play important roles in DNA repair at double-stranded breaks of nuclear DNA, both as long and small RNAs (*Ohle et al., 2016*; *Wei et al., 2012*). However, it is currently unclear whether RNA:DNA hybrids play a role in mitochondrial DNA recombination and repair (*Allkanjari and Baldock, 2021*).

Another possible explanation for the presence of small RNAs at block borders is that *T. gondii* simply utilizes all available genomic space for RNA production, including block borders. *T. gondii* appears to be as economical with its mitochondrial sequence space as other apicomplexans, and it does have hardly any unused noncoding sequences in its genome. In fact, the sequences at recombination sites could be regarded as an expansion of the mitochondrial genome sequence space, which is not available to other apicomplexans like the genus Plasmodium. There is evidence from recent polyA-sequencing efforts that long transcripts spanning several short RNAs are also produced in *T. gondii* (*Lee et al., 2021*). This is reminiscent of the situation in *P. falciparum*, where both strands of the entire mitochondrial genome are transcribed into polycistronic precursor RNAs. Therefore, block borders in *T. gondii* are likely transcribed into RNA as well. The functionality of these block border RNAs is unclear, but the unique sequence combinations generated at the block borders give rise to sRNAs that are not found in other apicomplexans with a simpler genome organization.

The discovery of sRNAs sharing sequences with mRNAs or being antisense to mRNAs in *T. gondii* mitochondria is an unexpected finding of our study that was facilitated by our improved understanding of mitochondrial genome organization. Antisense RNA is typically removed in mitochondria from flies and humans (*Pajak et al., 2019*). Similar antisense degradation mechanisms must be assumed in apicomplexans, as both strands of their genomes are transcribed in full (*Rehkopf et al., 2000*). The fact that three small antisense RNAs survive antisense RNA surveillance in *T. gondii* mitochondria is intriguing, and it remains to be established whether RNA:RNA interactions of these sRNAs with mRNAs occur in vivo and could be of regulatory impact.

Another potential regulatory link between mRNAs and sRNAs is represented by small RNAs that utilize coding regions. *RNA19* is of particular interest because it contains a piece of *coxIII* at the 5'-end and terminates close to the *cob* start codon, while also showing homology to rRNA. A few cases of mRNA sequences overlapping in antisense orientation with rRNA have been described in mammals and yeast (*Coelho et al., 2002*; *Kermekchiev and Ivanova, 2001*). Sequences homologous to rRNAs have been found in many coding regions in sense and antisense orientation (*Mauro and Edelman, 1997*). It has been suggested that this could link ribosome production to other cellular processes by reciprocal inhibition of mRNA and rRNA expression (*Coelho et al., 2002*). It is possible that *RNA19* and ribosome production could be balanced with *cob* protein production by tuning the processing or RNA degradation of *cob* mRNA. A quantitative analysis of RNA19 and *cob* mRNA accumulation under different conditions could help to clarify whether there is such an inverse correlation. Overall, the discovery of block-border sRNAs highlights the complex biogenesis of sRNAs in *T. gondii* mitochondria and will be a starting point to understand the processing of sRNAs and their function in general. Regarding their function, many, if not all, of the sRNAs at block borders could be used in ribosomes as rRNA fragments, which is discussed in the next chapter.

## sRNAs are incorporated into polysome-size complexes

A major unsolved problem in apicomplexan mitochondrial gene expression is the nature of the ribosome. Recently, compelling evidence has surfaced suggesting the existence of ribosome-sized particles, including ribosomal proteins. Using blue-native gel electrophoresis, it was demonstrated that tagged ribosomal proteins *Tg*mS35, *Tg*bL12m, *Tg*uL3m, and *Tg*uL24m migrate as constituents of a macromolecular complex within the size range of a ribosome (*Lacombe et al., 2019*; *Shikha et al., 2022*). Additionally, *Tg*mS35 and *Tg*uL24m coprecipitated with mitochondria-encoded rRNA. Knockdown of *Tg*mS35, *Tg*bL35m, *Tg*bL36m, and *Tg*bL28m resulted in a specific loss of activity of one respiratory complex, which is partially encoded in the mitochondrial genome (*Lacombe et al., 2019*; *Shikha et al., 2022*). Nevertheless, it remains to be determined whether these large complexes are actively involved in translation.

Here, we provide evidence that sRNAs with homology to rRNA are part of polysomes. This conclusion is based on our finding that sRNAs, mRNAs, and the ribosomal protein *Tg*uL11m are all found in high-molecular weight complexes that are sensitive to ribosome-dissociative conditions. We,

furthermore, observed a difference in the gradient distribution of sRNAs assigned to SSU and LSU, respectively. Based on a comparison with these two gradient distribution patterns, we were able to assign RNA29 to LSU. In sum, our findings strongly support the notion of active translation by mitochondrial ribosomes in tachyzoites of *T. gondii*, and demonstrate that most if not all mitochondrial small RNAs fragments are part of the apicomplexan mitochondrial ribosome. This includes sRNAs expressed from block borders like the mRNA sequence-containing RNA3 and RNA19. Also, block border RNAs RNA1 and RNA2 that share their 3'-end sequence, but differ in their 5'-part are both incorporated into ribosomes. RNA1 is of particular interest since it contains sequences homologous to two different parts of the LSU rRNA from *E. coli* and hence could occupy two positions within a single ribosome. RNA1 is almost exclusively found in ribosomes and polysomes, whereas RNA2 is also found free of ribosomes - similar to RNA29. This suggests that RNA2 and RNA29 might not integrate into ribosomes as effectively as other rRNA fragments, raising questions about the coordination of rRNA fragment assembly. The assembly of a ribosome with numerous short rRNA fragments and a comparable number of ribosomal proteins is undeniably complex. Notably, longer variants of RNA19 are not incorporated into ribosomes, unlike smaller isoforms, highlighting a connection between small RNA processing and ribosome assembly. In the case of RNA19, processing precedes integration, contrasting with plant chloroplast ribosomes where the 23 S rRNA is split endonucleolytically at two sites post-integration (*Liu et al., 2015*; *Nishimura et al., 2010*).

The analyses presented here mark a first step towards describing the assembly steps of the fascinating *T. gondii* ribosome. Unraveling the mechanics of ribosome assembly will require the identification of rRNA processing factors, which is within reach, given the excellent genetic toolbox available for *T. gondii*. It will also eventually require understanding the ribosome 3D structure. Within the spectrum of mitochondrial ribosome structures across species (*Ramrath et al., 2018*; *Saurer et al., 2023*; *Tobiasson et al., 2022*; *Waltz et al., 2021b*) mitoribosomes of *T. gondii* present an extreme example of ribosome diversification. The comprehensive collection of small RNA fragments identified in our study, including those at block borders, provides a valuable resource for future studies on the structure and function of the *T. gondii* mitochondrial ribosome.

## Conclusions

Apicomplexan mitochondrial genomes are vital for organism survival, expressing key respiratory chain and gene expression components, particularly ribosomes. The latter were proposed to be assembled from highly fragmented ribosomal RNAs, but whether rRNA fragments are expressed and used in ribosomes was unclear. We adapted a protocol to enrich *T. gondii* mitochondria (*Esseiva et al., 2004*) and used Nanopore sequencing to comprehensively map the genome with its repeated sequence blocks. Small RNA sequencing identified fragmented ribosomal RNAs, including some RNAs spanning block boundaries, thus fusing protein-coding and rRNA sequences. Sucrose density gradient analysis showed that such rRNA fragments are in polysome-size complexes. This distribution mirrored the localization of the mitoribosomal protein L11 as well as mRNAs in sucrose gradients. sRNAs, L11, and mRNAs shift to lower molecular weight complexes upon treatment with ribosome-dissociative buffer. We conclude that most if not all mitochondrial sRNAs are components of active ribosomes that assemble into polysomes. *T. gondii*'s dynamic block-based genome organization leads to the usage of mitochondrial sequences in mRNA as well as rRNA contexts, potentially linking rRNA and mRNA expression regulation.

## Methods
### Cultivation of host cells and *T. gondii* parasites

*T. gondii* parasites were cultured according to standard procedures ( *Jacot and Soldati-Favre, 2020*) unless otherwise indicated. Briefly, human foreskin fibroblasts (HFF-1, ATCC SCRC-1041) were grown as host cells in Dulbecco's modified Eagle's medium (DMEM) (Capricorn Scientific) supplemented with 10% fetal bovine serum (FBS) (Capricorn Scientific) and 100 µg/ml penicillin−streptomycin (Capricorn Scientific). *T. gondii* tachyzoites were maintained in flasks containing confluent HFF-1 cells in DMEM supplemented with 1% FBS and 100 µg/ml penicillin−streptomycin. *T. gondii* strains were kindly provided by Frank Seeber. Experiments were either carried out with parasites of the strain RH-ΔKUΔHX (*Huynh and Carruthers, 2009*) or RHpSAG1-βGal/pmt-GFP. The latter consists of RHβ1

(PMID: 8635747), which in addition expresses eGFP N-terminally fused to the *T. gondii* mitochondrial targeting sequence S9(33–159) (*DeRocher et al., 2000*).

## CRISPR/Cas9 mediated endogenous tagging of *Tgurpl11m*

C-terminal tagging of TGRH88_003980_t1 was performed as described previously (*Parker et al., 2019*). The sequence of a single guide RNA (sgRNA) specifically targeting the 3' end of TGRH88_003980_t1 was integrated into pSAG1:CAS9-GFPU6::sgUPRT (Addgene plasmid # 54467) using the Q5-site directed mutagenesis kit (NEB). The 3xHA epitope tag was amplified from the pPR2-HA3 plasmid (*Katris et al., 2014*) and integrated into the pU6-DHFR vector (Addgene plasmid #80329). The 3xHA-DHFR cassette was amplified from the resulting vector by Q5 polymerase (NEB) using primers containing a 50 nt overlap homologous to the either upstream or downstream regions of the TGRH88_003980_t1 stop codon (primers 'flank fwd *Tg*urpl11m tagging' and 'flank rev *Tg*urpl11m tagging' in *Supplementary file 9*, respectively). The amplified donor construct was together with the pSAG1::CAS9-GFPU6:sgUPRT plasmid encoding the Cas9 and the sgRNA co-transfected into RH-ΔKUΔHX as described previously (*Jacot and Soldati-Favre, 2020*). Parasites were grown under Pyrimethamine selection and subsequently cloned by limiting dilution in 96 well plates (*Roos et al., 1994*). Genotyping was performed directly from 96 well plates as described by *Piro et al., 2020*. Proper integration of the 3xHA tag was verified in single clones using Sanger Sequencing. Sequences of all the primers used for cloning and genotyping are listed in *Supplementary file 9*.

## Immunofluorescence assays

Immunofluorescence assays were done as described previously (*van Dooren et al., 2008*). Coverslips with confluent human foreskin fibroblasts (HFF) were infected with freshly egressed *Toxoplasma gondii* parasites. The next day, cells were fixed in 3% (w/v) paraformaldehyde in PBS for 15 min at room temperature and permeabilized in 0.25% (v/v) Triton X-100 in PBS for 10 min. Blocking was carried out in 2% (w/v) bovine serum albumin in PBS overnight. All antibody incubation steps were done at room temperature for 1 hr. Rabbit anti-*Tg*Tom40 (*van Dooren et al., 2016*, 1:2000 dilution) and rat anti-HA high affinity (Sigma, 11867431001, 1:200 dilution) primary antibodies were used. Then the coverslips were washed three times in PBS and incubated with donkey anti-rabbit CF 647 (Sigma, SAB4600177, 1:2000 dilution) and goat anti-rat AlexaFluor 488 (Thermo Fisher Scientific, A-11006, 1:500 dilution) secondary antibodies. Microscopy and image acquisition were performed on a DeltaVision Elite deconvolution setup (GE Healthcare) using an inverted Olympus IX71 microscope fitted with a UPlanSApo 100 x objective lens and Photometrics CoolSNAP HQ2 camera. Images were deconvolved using SoftWoRx Suite 2.0 software, brightness and contrast were linearly adjusted in FIJI/ImageJ (release 1.53 c).

## *T. gondii* organelle enrichment

A previously established protocol to enrich *T. gondii* organelles was modified here slightly (*Esseiva et al., 2004*). Freshly lysed *T. gondii* cultures from four T175 flasks (approximately $8 \times 10^8$ parasites) were filtered through a 3 µm pore size polycarbonate filter to remove host cell debris and harvested by centrifugation at $1500 \times g$ for 10 min. The parasite pellet was briefly washed in ice-cold Dulbecco's phosphate-buffered saline (PBS) (Capricorn Scientific) and again centrifuged for 10 min at $1500 \times g$. Afterwards, the parasite pellet was resuspended in 20 mM Tris-HCl, pH 7.4, 600 mM mannitol, 5 mM $MgCl_2$, and 2 mM EDTA containing 0.05% digitonin (Invitrogen) and incubated for 5 min on ice. The extract was clarified by centrifugation at $8000 \times g$ for 5 min at 4 °C. The supernatant containing soluble cytosolic components was discarded, and the corresponding pellet was resuspended in 20 mM Tris-HCl, pH 7.4, 600 mM mannitol, 5 mM $MgCl_2$, and 1 mM EDTA containing benzonase (345 U/ml) (Novagen), incubated for 20 min at room temperature and subsequently centrifuged at $8000 \times g$ for 5 min at 4 °C. The pellet was retained and washed in 20 mM Tris-HCl, pH 7.4, 600 mM mannitol, and 17 mM EDTA to inactivate benzonase remnants. Following a spin at $8000 \times g$ for 5 min at 4 °C, the pellet consisted of a crude enrichment of organelles and membranes. To study the *T. gondii* mitochondrial DNA content, the organelle pellet was directly used for DNA isolation. For organellar RNA analysis, the organelle pellet was frozen at –80 °C for at least 24 hr. Afterwards, it was thawed on ice and resuspended in a high detergent lysis buffer (20 mM HEPES-KOH pH 7.5, 100 mM KCl, 30 mM $MgCl_2$, 1% Igepal-CA630, 1.5% Triton X-100, 0.5% sodium deoxycholate, 1 mM DTT, EDTA-free protease

inhibitor). The lysate was incubated during continuous end-over-end rotation at 4 °C for 40 min and afterwards clarified by centrifugation at 16,000 × *g* for 10 min at 4 °C. The pellet, composed of cell debris and membranes, was discarded, and the supernatant, which contained the organelle lysate, was resuspended in TRIzol Reagent (Invitrogen) for RNA isolation.

## SDS–PAGE and immunoblotting

To detect mitochondrial GFP in RH pSAG1-βGal/pmt-GFP parasites, samples taken during organelle enrichment were directly diluted in Laemmli buffer (0.2 M Tris–HCl, 8% (w/v) SDS, 40% glycerol, 20% (v/v) β-mercaptoethanol, 0.005% bromophenol blue), incubated at 95 °C for 5 min and subjected to sodium dodecyl sulfate–polyacrylamide gel electrophoresis (SDS–PAGE). SDS–PAGE and immunoblotting were carried out as described previously (*Kupsch et al., 2012*). For immunodetection, monoclonal primary mouse anti-GFP antibody (G1546, Sigma–Aldrich) and polyclonal rabbit anti-heat shock protein 70 antibody (AS05 083 A, Agrisera) followed by secondary horseradish peroxidase (HRP)labeled goat antimouse antibody (ab205719, Abcam) and goat anti-rabbit antibody (ab205718, Abcam) were used.

Proteins from sucrose gradient fractions were isolated using methanol-chloroform-water extraction (*Wessel and Flügge, 1984*). Protein pellets were resuspended in Laemmli buffer, heated for 5 min at 95 °C, and subjected to SDS-PAGE and immunoblotting. Primary rat anti-HA antibody (11867423001, Roche) and secondary rabbit anti-rat (HRP) antibody (ab6734, Abcam) were used for *Tg*uL11m:HA detection.

## DNA isolation and library construction

DNA was isolated from organelle-enriched fractions using the GeneMATRIX Tissue DNA Purification Kit (Roboklon) according to the manufacturer's protocol for cultured cells. The Oxford Nanopore Technology sequencing library was prepared with 1 µg of organelle-enriched DNA using the SQK-LSK108 ligation sequencing kit (Oxford Nanopore, ONT) according to the manufacturer's instructions. Sequencing was performed on an R9.4.1 Spot ON Flow Cell; live-basecalling was done using the ONT Guppy software package.

## RNA isolation and RNA-seq library construction

RNA was isolated from total organelle enriched and sucrose gradient fraction samples using TRIzol Reagent (Invitrogen), followed by the Monarch Total RNA Miniprep Kit (New England BioLabs) applying the manufacturer's protocol for TRIzol extracted samples. Two hundred nanograms of RNA was used to generate sequencing libraries using the NEBNext Multiplex Small RNA Library Prep Set for Illumina Kit (New England BioLabs) according to the manufacturer's instructions. For identification of PCR duplicates in the library preparation process, the 5'-linker was modified with eight N-nucleotides (as unique molecular identifiers, UMIs). Small RNA library clean-up was carried out with AMPure XP Beads (Beckmann Coulter). Sequencing was performed by Genewiz on an Illumina NovaSeq platform with a read length of 150 bp for each pair.

## Agarose-formaldehyde gel electrophoresis

RNA samples were diluted in denaturing loading buffer (Deionized formamide 62.5% (v/v), formaldehyde 1.14 M, bromophenol blue 200 µg/mL, xylene cyanole 200 µg/mL, MOPS-EDTA-sodium acetate), and separated on a 1% agarose gel containing 1.2% formaldehyde. Cytosolic rRNAs were detected with ethidium bromide staining.

## Denaturing urea-PAGE and sRNA gel blot

RNA was separated by denaturing Urea-PAGE (10% or 12% polyacrylamide gel for total RNA/organelle enriched RNA and RNA extracted from sucrose density gradient fractions, respectively). Small RNA gel blotting and detection of transcripts with radiolabeled DNA oligo probes was carried out as described previously (*Loizeau et al., 2014*). Sequences of the oligo probes used in this study are listed in *Supplementary file 9*. Blot hybridization of sucrose density gradient fractions was carried out consecutively in the order RNA1_2, RNA19, SSUA, LSUD_E, RNA3, LSUF_G, RNA29, with removal of probes by heating the blot to 60 °C for 1 hr in 0.5% SDS.

## 5' RACE

*T. gondii* total RNA was ligated to a small RNA oligo (Rumsh, see *Supplementary file 9*) using T4 RNA Ligase 1 (NEB) according to the manufacturer's instructions. Afterwards, RNA was reverse transcribed using ProtoScript II Reverse Transcriptase (NEB) and random primers. PCR amplification of cDNA was performed with a gene-specific *cob* primer and a primer annealing to the ligated oligo (Rumsh1, see *Supplementary file 9*) using Taq DNA Polymerase (Roboklon). The purified PCR product was sequenced by Sanger Sequencing (LGC Genomics GmbH).

## Sucrose density gradient centrifugation

Ribosome fractionation by sucrose density gradient centrifugation was performed as previously described *Waltz et al., 2021a*. Briefly, *T. gondii* parasites were harvested and organelle enrichment was performed as described above with small modifications. Salt concentration in the digitonin and lysis buffer was either adjusted for optimal mitoribosome stability (30 mM $MgCl_2$, 100 mM KCl) or for dissociative conditions (10 mM EDTA, 300 mM KCl). The organelle-enriched lysate was layered onto a continuous 10–50% sucrose gradient. Sucrose solutions (10% and 50%) were previously prepared freshly in 20 mM HEPES-KOH pH 7.5,1 mM DTT, EDTA-free protease inhibitor, and either 30 mM MgCl2, 100 mM KCl or 10 mM EDTA, 300 mM KCl (dissociative conditions). A continuous sucrose gradient was formed using BioComp Gradient Master 108. The gradient was centrifuged at 200.000 × g for 2 hr at 4 °C in a S52-ST rotor (Thermo Scientific) and subsequently fractionated into 14 fractions using the BioComp Piston Gradient Fractionator device. Fractions were resuspended directly in TRIzol reagent (Invitrogen). For RT-qPCR samples 1.2 ng of human estrogen receptor 1 (ESR1) RNA was introduced into each fraction as spike-in control. RNA was extracted from fractions using Trizol-Chloroform phase separation followed by overnight precipitation with 1 volume Isopropanol.

## RT-qPCR

RNA isolated from fractions was treated with TurboDNase (NEB) and afterwards subjected to reverse transcription using ProtoScript II Reverse Transcriptase (NEB). Synthesized cDNA from each fraction was used as a template in qPCR. Reactions were performed as technical triplicates using Luna Universal qPCR Master Mix (NEB). Primers listed in *Supplementary file 9* were used for the amplification of mitochondrial cob, coxI, coxIII, LSUF/G rRNA, and ESR1 spike-in for normalization. The relative quantity of each amplicon was determined using the Pfaffl method (*Pfaffl, 2001*). Relative quantities for amplicons in each fraction are presented as a percentage of the cumulative relative quantities of the respective amplicon across all fractions. Data was visualized using GraphPad Prism Version 10.1.0.

## Bioinformatic analyses

All bioinformatic analyses except for the quantification of sequence block combinations were performed on the Galaxy web platform (v.22.01; https://usegalaxy.eu, *Afgan et al., 2018*). All reference sequences used were retrieved from GenBank (*Sayers et al., 2019*). Sequence alignments were prepared using Clustal Omega (https://www.ebi.ac.uk/Tools/msa/clustalo/) with default settings. *E. coli* sequences were added manually according to sequence positions established previously (*Feagin et al., 2012*).

## Masking NUMTs in the *T. gondii* nuclear genome

To identify NUMTs in the *T. gondii* nuclear genome, mitochondrial sequence blocks (MN077088.1-MN077111.1) were aligned against the RH-88 nuclear reference genome (GCA_019455545.1, August 2021) using NCBI-BLASTN (v. 2.13.0) (*Altschul et al., 1990*) (default parameters, except max_target_seqs = 5000). All hits obtained in the BLASTN tabular output were manually integrated into a GFF3 file format. NUMTs in the nuclear genome were masked with bedtools MaskFastaBed (*Quinlan and Hall, 2010*) based on the intervals defined in the GFF3 file. In total, we masked 8118 sites with an average length of 92 nt representing in total ~1% of the nuclear genome.

## DNA sequencing data analysis

DNA Oxford Nanopore sequencing data were trimmed using Porechop (Version 0.2.4, https://github.com/rrwick/Porechop; *Wick, 2018*). As a first step, trimmed reads were mapped against the human reference genome GRCh38.p14 (GCA_000001405.29, February 2022) using Minimap2

(Version 2.24) (*Li, 2018*)(PacBio/Oxford Nanopore read to reference mapping, default settings). Mapping results were filtered using SAMtools view (Version 1.15.1) (*Li et al., 2009*) (-f read is unmapped). Unmapped reads were subsequently simultaneously mapped against the *T. gondii* RH-88 nuclear reference genome masked for NUMTs, the apicoplast genome (CM033583.1), and mitochondrial sequence blocks (MN077088.1- MN077111.1) using Minimap2. A fraction of the reads that fit best to the mitochondrial genome still retain shorter nuclear sequence sections and are, therefore, representing NUMTs. We, therefore, took all reads that mapped to the mitochondrial genome in our first mapping and mapped them against the nuclear reference genome masked for NUMTs using Minimap2 (Version 2.24). The remaining unmapped reads were considered mitochondrial and visualized with Geneious Prime v. 2019.2.3. (Biomatters Ltd.). Reads were annotated providing the mitochondrial sequence blocks (MN077088.1 - MN077111.1) and ORFs (MN077082.1 - MN077084.1) as annotation database using the Geneious Prime 'Annotate from Database' tool with similarity set to ≥75%.

Sequence comparisons of ONT reads found here with published ONT reads for the *T. gondii* mitochondrial genome were performed using the NUCmer tool in the MUMmer package of Galaxy with the following settings: minimum match length 20 and minimum alignment length 290 (shortest read in *Namasivayam et al., 2021*). The Show-Coords tool was used to output alignment information. We found 1555 reads of our dataset being entirely part (defined as ≥99% read coverage) of reads from the *Namasivayam et al., 2021* and 212 reads of their dataset being fully found within reads from our dataset.

## Analysis of mitochondrial DNA sequence block arrangements

For a comprehensive overview of the analysis, including the complete.Rmd report please refer to the GitHub repository (https://github.com/Kovox91/ToxoBlocks, copy archived at *Maschmann, 2024*). In brief, annotated block IDs, indicating the directionality of each block, were exported in plain text. A custom R script, leveraging the tidyverse library (*Wickham et al., 2019*), was employed for the subsequent analysis. This script facilitated the removal of block combinations where two blocks were distanced by more than 10 nucleotides. For two-block combinations, the script summed up all identified combinations, filtering out those occurring less than 50 times. The results, incorporating block orientations, were visualized using ComplexHeatmap (*Gu and Hübschmann, 2022*). To quantify the relative abundance of ORFs, counts for the respective ORF combinations were tallied across all reads. The ORF frequency was then computed as the ratio of reads containing at least one instance of the specific ORF to the total number of reads that were at least as long as the given ORF. RNA sequencing data analysis.

Adapter and quality trimming of RNA sequencing data was performed using Cutadapt (Version 4.1; *Martin, 2011*). Umi-tools was used to extract UMIs (Version 1.1.2; *Smith et al., 2017*). Reads were aligned to the *T. gondii* RH-88 nuclear reference genome masked for NUMTs and the apicoplast genome (CM033583.1) using Bowtie2 (Version 2.4.5; *Langmead et al., 2009*) in default settings. Mapping results were filtered using SAMtools view (Version 1.15.1; *Li et al., 2009*) (-f read is unmapped). Unmapped reads were kept and mapped with Bowtie2 using default settings against the *T. gondii* mitochondrial genome using all combinations of the sequence blocks found in our genome sequence analysis (GenBank accession MN077088.1 - MN077111.1, OR086910 - OR086916) as determined in our ONT DNA sequencing data. Transcripts were determined as follows: first, most transcripts had short poly-A tails - up to 12 adenines at their 3'-ends - which is similar to the situation in *P. falciparum* (*Rehkopf et al., 2000*). For the identification of the transcript 3'-ends, all polyA-tails longer than three nucleotides were removed from reads using fastp (*Chen et al., 2018*). The number of mapped 3' and 5' read ends at each position in the reference was calculated using Bedtools Genome Coverage 2.30.0 (*Quinlan and Hall, 2010*). The 5'-ends at the beginning of coverage plateaus over the background were considered starting points of transcripts. For each 5' end, the mapped 3'-end most far away but still represented by at least 10% of the reads of the 5'-terminus and connected to the starting point by uninterrupted coverage over the background was considered the corresponding 3'-terminus. Alignments and coverage graphs were visualized in Integrated Genome Viewer (*Robinson et al., 2011*). All RNA sequencing data are available at SRA under accession number PRJNA978626.

## Sequence similarity search

All newly identified transcripts were subjected to an NCBI-BLASTN homology search (v. 2.13.0; *Altschul et al., 1990*; default parameters, except word size = 7) against the *Plasmodium falciparum* mitochondrial genome (M76611.1) and the *Eimeria leuckarti* mitochondrial genome (MW354691.1). Transcripts not matching the *P. falciparum* or *E. leuckarti* mitochondrial genome were further inspected for homology to the *E. coli* rRNA operon (J01695.2) using BLASTN.

## Data access

The RNA and DNA sequencing data generated in this study have been submitted to the NCBI BioProject database (https://www.ncbi.nlm.nih.gov/bioproject/) under accession number PRJNA978626. New block annotations can be found in GenBank, OR086910 - OR086916.

## Acknowledgements

We are grateful to Giel van Dooren, Frank Seeber, Kai Matuschewski, and Hannes Ruwe for the critical discussion of our data and manuscript. We sincerely thank Frank Seeber for providing the *T. gondii* strains. We are grateful to Giel van Dooren for providing the plasmids for CRISPR/Cas9 tagging and for support from the microscopy facility at ANU Canberra. We thank Svea Beier for preparing the constructs for RPL11 tagging and Florian Rösch for preparing the vector for the ESR1-spike in amplification. This work was supported by the German research foundation (DFG) by grant IRTG2290, project B01 to CS. The article processing charge was funded by the Deutsche Forschungsgemeinschaft (DFG, German Research Foundation) – 491192747 and the Open Access Publication Fund of Humboldt-Universität zu Berlin.

## Additional information

### Funding

| Funder | Grant reference number | Author |
| --- | --- | --- |
| Deutsche Forschungsgemeinschaft | IRTG2290-B01 | Christian Schmitz-Linneweber |

The funders had no role in study design, data collection and interpretation, or the decision to submit the work for publication.

### Author contributions

Sabrina Tetzlaff, Arne Hillebrand, Formal analysis, Investigation, Methodology, Writing - review and editing; Nikiforos Drakoulis, Validation, Investigation, Writing - review and editing; Zala Gluhic, Investigation; Sascha Maschmann, Resources, Data curation, Formal analysis, Writing - review and editing; Peter Lyko, Formal analysis, Investigation; Susann Wicke, Data curation, Formal analysis, Supervision, Writing - review and editing; Christian Schmitz-Linneweber, Conceptualization, Supervision, Writing - original draft, Writing - review and editing

### Author ORCIDs

Sabrina Tetzlaff http://orcid.org/0009-0009-1786-2890
Sascha Maschmann http://orcid.org/0000-0002-0560-9021
Susann Wicke http://orcid.org/0000-0001-5785-9500
Christian Schmitz-Linneweber http://orcid.org/0000-0002-6125-4253

### Decision letter and Author response

Decision letter https://doi.org/10.7554/eLife.95407.sa1
Author response https://doi.org/10.7554/eLife.95407.sa2

# Additional files

## Supplementary files

- Supplementary file 1. Oxford Nanopore sequencing technology (ONT) sequencing mapping statistics on nuclear and organellar genomes.
- Supplementary file 2. Read length distribution of *T. gondii* mitochondrial Oxford Nanopore sequencing technology (ONT) reads.
- Supplementary file 3. Mitochondrial sequence block frequencies in Oxford Nanopore sequencing technology (ONT) DNA sequencing data.
- Supplementary file 4. Sequence block combinations identified in *T. gondii* mitochondrial Oxford Nanopore sequencing technology (ONT) reads using a custom R script. Combinations that were found less than 50 times are considered false positives and shown in gray.
- Supplementary file 5. Fraction of reads containing full-length open reading frames.
- Supplementary file 6. Mapping statistics of the RNA-seq data on the different genomes/sequence blocks. After filtering the raw reads against the nuclear rRNA genes, the remaining reads were mapped against the three subgenomes of *T. gondii* RH-88.
- Supplementary file 7. List of mitochondrial non-coding RNAs identified by small RNA (sRNA) sequencing.
- Supplementary file 8. Overview of mitochondrial non-coding RNAs identified in *P. falciparum*.
- Supplementary file 9. List of oligonucleotides used in this study.
- MDAR checklist

## Data availability

The RNA and DNA sequencing data generated in this study have been submitted to the NCBI BioProject database (https://www.ncbi.nlm.nih.gov/bioproject/) under accession number PRJNA978626. New block annotations can be found in GenBank, OR086910–OR086916.

The following datasets were generated:

| Author(s) | Year | Dataset title | Dataset URL | Database and Identifier |
|---|---|---|---|---|
| Tetzlaff S, Schmitz-Linneweber C | 2023 | Characterization of short RNAs, in particular rRNAs from mitochondria of *Toxoplasma gondii* | https://www.ncbi.nlm.nih.gov/bioproject/?term=PRJNA978626 | NCBI BioProject, PRJNA978626 |
| Schmitz-Linneweber C, Tetzlaff S | 2024 | *Toxoplasma gondii* element block F genomic sequence; mitochondrial | https://www.ncbi.nlm.nih.gov/nuccore/OR086910 | NCBI Nucleotide, OR086910 |
| Schmitz-Linneweber C, Tetzlaff S | 2024 | *Toxoplasma gondii* element block K genomic sequence; mitochondrial, revised | https://www.ncbi.nlm.nih.gov/nuccore/OR086911 | NCBI Nucleotide, OR086911 |
| Schmitz-Linneweber C, Tetzlaff S | 2024 | *Toxoplasma gondii* element block M genomic sequence; mitochondrial, revised | https://www.ncbi.nlm.nih.gov/nuccore/OR086912 | NCBI Nucleotide, OR086912 |
| Schmitz-Linneweber C, Tetzlaff S | 2024 | *Toxoplasma gondii* element block H genomic sequence; mitochondrial, revised | https://www.ncbi.nlm.nih.gov/nuccore/OR086913 | NCBI Nucleotide, OR086913 |
| Schmitz-Linneweber C, Tetzlaff S | 2024 | *Toxoplasma gondii* element block C genomic sequence; mitochondrial, revised | https://www.ncbi.nlm.nih.gov/nuccore/OR086914 | NCBI Nucleotide, OR086914 |
| Schmitz-Linneweber C, Tetzlaff S | 2024 | *Toxoplasma gondii* element block Q genomic sequence; mitochondrial, revised | https://www.ncbi.nlm.nih.gov/nuccore/OR086915 | NCBI Nucleotide, OR086915 |
| Schmitz-Linneweber C, Tetzlaff S | 2024 | *Toxoplasma gondii* element block B genomic sequence; mitochondrial, revised | https://www.ncbi.nlm.nih.gov/nuccore/OR086916 | NCBI Nucleotide, OR086916 |

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
