## [Editor Report]

This study brings a solid methodological advancement on previous attempts to understand the nature of the mitochondrial genome in *Toxoplasma*, applicable to Apicomplexa in general. The authors achieve extensive mitochondrial sequencing through a compelling and validated methodology, providing valuable RNA data. The improved catalog of mitochondrial rRNA and identification of overlapping protein-coding and rRNA genes will interest researchers in mitochondrial evolutionary biology and the mitoribosome community.

---

## [Decision Letter]

[Editors' note: this paper was reviewed by Review Commons.]

---

## [Author Response]

General Statements

Thank you for providing us the opportunity to submit a revised version of our manuscript. To address the reviewers' concerns, we have significantly enhanced the manuscript by adding extensive data. This new data is crucial for demonstrating that our polysome analysis can effectively differentiate between polysomes and monosomes. It strengthens our key conclusion that the mitochondrial sRNAs we detected by a combination of nanopore-based genome sequencing and small RNA sequencing are used in actively translating ribosomes. This is to date the most direct evidence for translation in apicomplexan mitochondria. Moreover, the small RNAs generated from genomic recombination sites that are fusions of mRNA and rRNA sequences are found in polysomes as well. As far as we know, the linking of rRNA production to genomic recombination sites has not been described before.

The data added are:

We introduced an additional transgenic line of *T. gondii* that produces a tagged variant of the ribosomal protein L11. This step was crucial to illustrate the distribution of ribosomes in our sucrose gradient analysis.We examined the distribution of all three mitochondrial mRNAs in our sucrose gradient centrifugation experiments, showing that they are present in large molecular weight complexes susceptible to ribosome dissociation conditions.Our results indicate that small RNAs mirror the behavior of mRNAs and L11 in our gradients, suggesting their role in the formation of ribosomes and, subsequently, polysomes.

The generation of these data involved Zala Gluhic, who conducted the IFA of the tagged L11 line, Nikiforos Drakoulis who carried out the mRNA analysis in sucrose gradients, and Sascha Maschmann, who re-evaluated some of our mitochondrial genome Nanopore sequencing data bioinformatically, as per the reviewers' request. Consequently, all three contributors are now coauthors of the manuscript.

Point-by-point description of the revisionsReviewer #1

Your appreciation of our work is greatly encouraging; thank you for analyzing our manuscript!

Evidence, reproducibility and claritySummary:Mitochondrial genomes of Apicomplexa parasites have undergone dramatic reductions during their evolution with genes for only three proteins remaining. In addition, ribosomal RNA genes are present in different, often species-specific gene arrangements. *Toxoplasma* exhibits massive variations in gene arrangement that are distributed over multiple copies. In this study the Schmitz-Linneweber lab not only re-analysed the mitochondrial genome of *Toxoplasma gondii* using a novel protocol for enriching the organellar nucleic acid, allowing to sequence the mitochondrial genome at unprecedented depth, they also addressed an enigma regarding the expression status of mitochondrial ribosomes.While indirect evidence of mitochondrial translation exists, no direct evidence for active mitoribosomes exist and their composition is still poorly understood. Here, using HTS or small RNAs the authors demonstrate that they are incorporated into polysomes. Furthermore, the authors developed the hypothesis that the block-based genome organization enables the dual utilization of mitochondrial sequences as both messenger RNAs and ribosomal RNAs.Own opinion/Major commentsThe mitochondria of the Apicomplexa are characterized by massive gene transfer into the cell nucleus, and sequence rearrangements, which has led to a single, questioned genome reorganization. The underlying mechanisms of gene transcription and translation are also poorly understood. In a previous study, the Kissinger lab demonstrate the unique organization of the mitochondrial genome that consists of minimally of 21 sequence blocks (SBs) totaling 5.9 kb that exist as nonrandom concatemers (Namasivayam et al. 2021).In this study the authors optimized a new isolation technique of organellar content to sequence the mitochondrial genome. This new purification protocol appears to be very robust and allowed the sequencing of mitochondrial genome at unprecedented depth. The obtained data not only validate previous studies, but they also suggest several new features, such as (potentially) continuous reshuffling of DNA blocks, leading to independent block combinations.The most important aspect of this study is the demonstration of polysomes and the presence of rRNAs within these complexes, taking previous studies (i.e. Lacombe et al., 2019) a step further.Taking all these efforts and data into account it is a very nice and interesting study that will certainly be of interest for a broader readership. All the presented data and analysis appear to be solid and well controlled. However, it must be mentioned that this reviewer is not an expert when it comes to the analysis and comparison of huge genomic datasets and the opinion of a bioinformatician would be helpful in assessing this study in more detail.All other data (organellar purification and analysis of polysomes) appear state of the art and no corrections are required.SignificanceGeneral assessment:Taking all these efforts and data into account it is a very nice and interesting study that will certainly be of interest for a broader readership. All the presented data and analysis appear to be solid and well controlled. However, it must be mentioned that this reviewer is not an expert when it comes to the analysis and comparison of huge genomic datasets and the opinion of a bioinformatician would be helpful in assessing this study in more detail.All other data (organellar purification and analysis of polysomes) appear state of the art and no corrections are required.Advance:The study fills an important gap in our knowledge regarding the organization and translational activity of the apicomplexan (*Toxoplasma*) mitoribosome. See also comments above.Audience: Cell Biology, Parasitology, MitochondriaReviewer #2

We sincerely thank you for your constructive feedback and the thorough evaluation of our manuscript.

Evidence, reproducibility and clarityIn this article, the authors delve into an intriguing topic, aiming to enhance our understanding of the organization of the mitochondrial genome of *T. gondii*, a parasite of significant importance in both human and animal health contexts.In essence, their approach involves enriching mitochondrial material, followed by genome sequencing and the analysis of mitochondrial short RNAs. They achieve a remarkable depth of mitochondrial sequencing and generate valuable RNA data. Furthermore, their efforts lead to the discovery and annotation of new short RNAs.Overall, the article is well-crafted and presents compelling results. However, it's worth noting that, at times, the authors appear somewhat self-congratulatory, and certain results might be perceived as overly ambitious. Nevertheless, the discussion is aptly constructed.

Major comment: (we have numbered the comments of the reviewer)

1. They assert certain discoveries that had already been reported. Notably, they adapt an existing protocol for mitochondrial enrichment and describe it as 'We developed a protocol to enrich *T. gondii* mitochondria.'

We did reference the protocol that our method was built on in the methods section: “A previously established protocol to enrich *T. gondii* organelles was modified here slightly (Esseiva et al. 2004).” We agree that the phrasing “developed a protocol” in the Conclusions chapter is an overstatement and changes this to “adapted a protocol, citing Esseiva et al. (line 675)

2. It's worth noting that they neither reference a more recently described protocol (PMC6851545) nor compare the performance of their modified protocol with the original.

We added this reference, but since we have not been using the Lacombe protocol mentioned by the reviewer and have no need to compare efficiencies to support our conclusions (this not being a methods paper), we would rather abstain from a protocol comparison / citation. We do add info on why our protocol was serving our DNA-sequencing efforts very well (see next answer please).

3. The protocol they employ does not seem to yield exceptionally high success rates, as mitochondrial DNA constitutes less than 10% of the total sequenced DNA.

We value the reviewer's critical evaluation. We understand that the proportion of mitochondrial sequences in our data may imply a low success rate. Nevertheless, we would suggest a comparison of the proportion of mitochondrial DNA sequenced from a total DNA sample (Namasivayam et al.) versus DNA obtained from the organelle enrichment protocol (Supplementary file 1). With the organelle enriched sample we could obtain a ~42-fold increase in mitochondrial sequencing depth. When compared to the RH Δuprt ONT sequencing data from Namasivayam et al. the increase is even ~184-fold.

We have added following info to the text:

Line 177-179:

“This is a 42-fold increase in the sequencing depth of the *T. gondii* mitochondrial genome compared to previous attempts (Supplementary file 1), which can be attributed to the effectiveness of the purification process.”

4. Additionally, they frequently mention the identification of specific combinations of sequence blocks previously identified by Namasivayam et al. (PMC8092004), which was also discussed in Namasivayam et al. 2021."

After publishing this manuscript on bioRxiv, we were approached by an author from the Namasivayam paper, who also raised this point and helped us to improve the manuscript substantially and to properly represent previous work by the Kissinger lab. We apologize for this failure to properly represent important previous work. According to your and this authors’ advice, we amended the text at several positions:

Lines 35-38 (abstract):

“It has been established previously that the *T. gondii* genome comprises 21 sequence blocks that undergo recombination among themselves, but that their order is not entirely random. The enhanced coverage of the mito genome allowed us to characterize block combinations at increased resolution.”

Line 115: we added the citation:

“The *T. gondii* nuclear genome contains many insertions of mitochondrial DNA sequences (Ossorio et al. 1991; Gjerde 2013; Namasivayam et al. 2023)”

Line 169-169: we deleted the following sentence:

“However, a quantitative analysis of block combinations remained beyond reach due to the low amount of sequence reads.”

Line 191: we added the following sentence:

*“,* thus corroborating at a deeper sequencing coverage conclusions so far gained from a more limited read set (Namasivayam et al. 2021; Berná et al. 2021a).”

Line 200: we added a sentence to specify the work by Namasivayam versus our work:

“, confirming block combinations observed previously (Namasivayam et al. 2021) and adding combination frequencies based on higher read numbers.”

Line 222:

“Full-length coding regions had been found previously on nanopore reads in *T. gondii* (Namasivayam et al. 2021). In our improved representation of the mitogenome, we identified a large number of full representations of *cob*, *coxI* and *coxIII* in our dataset […].”

Line 221: Namasivayam et al. did not claim post-transcrional shuffling of blocks as we erroneously stated; thus we deleted their reference here and added a sentence to make clear that we use approaches established by Namasivayam et al. for describing block combinations.

Now line 227:

“This may be adequate for the expression of the encoded proteins; however, we cannot currently exclude the possibility of genomic or post-transcriptional block shuffling that could lead to more complete open reading frames (as discussed in Berná et al., 2021b). We next applied approaches established previously to represent biased recombination events based on alternative block combinations (see Figure S12 in Namasivayam et al. 2021) to our improved ONT read set.”

Line 278 and 284.

Namasivayam et al. did not annotate but rather predict rRNAs based on homology searches. We changed the verb accordingly.

Now line 267-268:

“Among these, 11 correspond to previously predicted rRNA fragments (Namasivayam et al., 2021).”

Now line 271-273:

“This included a reassignment of SSUF to the opposite strand and also affected the four rRNA fragments LSUF, LSUG, LSUD, and LSUE, which had been predicted as separate transcripts of the large subunit (LSU) of the ribosome (Namasivayam et al., 2021).”

Now line 305-308:

“We next analyzed the accumulation of selected RNAs previously undetected in *T. gondii* for the sequence blocks Kp-K: Transcripts RNA5, 17, and 29, as well as the already predicted transcripts RNA10 and SSUD are found on the minus strand (strandedness according to Namasivayam et al., 2021; MN077088.1 – MN077111.1).”

Line 567:

The individual ONT reads are not repeated in the two previously published datasets (discussed in Berná et al., 2021b) and we also did not find overlap between our data presented here and published previously (Namasivayam et al., 2021). Thus, the large number of block combinations identified here reinforces and elevates previous conclusions that continuous recombination shuffles the blocks (Namasivayam et al. 2021).”

5. Missing in the supplementary material are basic details on the sequences performed. Distribution of mitochondrial reads length, depth, etc.

We included a supplemental table (Supplementary file 2) showing read length distribution of mitochondrial ONT reads. Other basic information on ONT sequencing and Illumina sRNA sequencing data we provide in Tab S1 and S6.

6. Further clarification is needed for Figure 2. Specifically, the frequency units or combinations of frequency (A, B, and C) are not clearly explained. While the matrix's asymmetry suggests a 5'- 3' orientation difference, this orientation difference is not explicitly specified (B). Additionally, the fragment Mp does not appear in the block combination figure (C).

We thank the Reviewer for pointing out the lack of clarity in Figure 2. In Figure 2 we show the number of occurrences of individual blocks- and block combinations across the entire set of mitochondrial ONT reads. To enhance clarity we revised the figure caption and included additional details. We have moved Figure 2B to the supplementary section (now Figure 2—figure supplement 1A), because the information shown there is mostly redundant with Figure 2C. We refined the R code for analyzing the block combinations by also considering block directionality. Furthermore we removed combinations of blocks with gaps or overlaps between the respective annotated blocks larger than 10 nucleotides. This removed the majority of lowly abundant block combinations. The heatmap presenting the results is now differently labeled and includes a legend for distinguishing between block orientations. Mp was merged to fragment B since no independent B fragment was observed in the dataset. We added a sentence in the capture of Figure 2—figure supplement 2 explaining the reasoning behind merging block B and Mp. The new sequence version was published in GenBank OR086910-OR086916.

Some points to improve the introduction:7. Provide an evolutionary context for the following phrase: 'An idiosyncratic feature of Apicomplexa is a highly derived mitochondrial genome.' Specify what you intend to emphasize.

Line 55. We deleted this sentence since this is explained further down in greater detail (now line 61):

Dozens of apicomplexan mitochondrial genomes have been sequenced (Berná et al. 2021b). These sequences showcase the extreme reductive evolution in apicomplexan mitochondria, setting records for the smallest mitochondrial genomes known to date, ranging in length from 6 to 11 kb (Hikosaka et al. 2013; Oborník and Lukeš 2015).”

Line 56: The sentence must begin with a capital letter

OK, thanks.

8. In line 58 "Nuclear genes encoding proteins with functions in mitochondria contribute strongly to *P. falciparum* and *T. gondii* cell fitness"Although it is mentioned later, it would be more effective to introduce the fact that all but three genes are encoded in the nucleus.

We added the following:

Line 59-61:

“With only three exceptions, all mitochondrial proteins are nuclear encoded and these nuclear genes contribute strongly to *P. berghei* and *T. gondii* cell fitness (Sidik et al. 2016; Bushell et al. 2017).”

9. Line68: "Apicomplexan mitogenomes usually code only for three proteins"It seems to me that 'usually' should not be included.

Agreed.

10. Line 65-67: The sentence should include that the mitochondrial genome is composed of a total of 20 blocks of repeating sequences organized in multiple DNA molecules of varying length and non-random combinations

The corrected sentence reads (line 67-69):

“The *T. gondii* mitochondrial genome is composed of 21 repetitive sequence blocks that are organized on multiple DNA molecules of varying lengths and non-random combinations (Namasivayam et al. 2021; Berná et al. 2021a).”

11. At the end of the introduction, the authors state that they have developed a protocol for mitochondrial enrichment. The text should be modified taking into account that:1- The new protocol is an adaptation of another existing protocol. In fact, the Methods the authors say the protocol was "slightly" modified.

We rephrased the sentence (line 105-107):

“We used a slightly modified protocol of mitochondria enrichment (Esseiva et al. 2004) to investigate the structure of the mitochondrial genome through long-read sequencing at an unprecedented depth.”

2- There is already existing mitochondrial enrichment protocol available [Reference: https://www.ncbi.nlm.nih.gov/pmc/articles/PMC6851545/#mmi14357-bib-0074]. In any case, they should consider performing a comparative analysis between the proposed protocol and existing ones to determine its relative effectiveness. It should be noted that the proposed protocol enriches in organelles (including the nucleus and apicoplast), but when sequencing DNA, mitochondrial DNA accounts for only 5% of the total reads, which may raise doubts about its overall efficacy.

Thank you for pointing out this protocol. Since we based our protocol on Esseiva et al., we will not cite Lacombe et al. here. The modified Esseiva protocol allowed a 42 fold better coverage of mitochondrial genome sequences over previous efforts, which is all we asked for. We do not intend to focus here on methods development and comparisons and clearly should not have stressed novelty as we did. The method is now appropriately described and referenced in abstract, introduction, and results. Please see also our answer to your point #2.

Benzonase is 60 kDa and will move into the nucleus. And yes, the apicoplast will be co-enriched with mitochondria, which is however not an issue given the good coverage we get for mitochondrial DNA (and we wrote “leaving mitochondria and other organelles intact”).

Some points related to Results section:12. Lines 113-115: 'To distinguish between NUMTs (nuclear DNA sequences that originated from mitochondria) and true mitochondrial sequences, it is necessary to enrich mitochondrial DNA.' I disagree with this sentence. NUMTs, in general, consist of very short sequences. With long reads, it is relatively straightforward to differentiate mitochondrial sequences from those nuclear sequences that have small mitochondrial fractions. In my opinion, even many Illumina reads can be confidently identified as belonging solely to the mitochondria. I found this article that supports this argument, indicating that the majority of NUMTs are less than 100 nucleotides in length [Reference: https://pubmed.ncbi.nlm.nih.gov/37293002/].

According to the paper you cite, there are hundreds of NUMTs that are longer than 100 nt and thus pose potential problems for mis-mapping of Illumina reads. We believe it is therefore helpful to increase sequencing depth using enrichment of mitochondria and long reads to say with high confidence that the sequences are of mitochondrial origin. We are however agreeing that toning down our statement is appropriate given that Illumina read alone are also already allowing analyses of mitochondrial DNA long read sequencing. The phase now reads like this (line 116-118):

“To better distinguish between NUMTs (nuclear DNA sequences that originated from mitochondria) and true mitochondrial sequences, it is helpful to enrich mitochondrial DNA.”

13. Lines 166-168: 'A previous sequencing study used Oxford Nanopore sequencing technology (ONT) to identify combinations of sequence blocks in *T. gondii* mitochondria (Namasivayam et al. 2021).' However, it's important to note that Namasivayam's group did not merely use ONT to identify combinations of blocks; rather, they discovered, identified, and defined these combinations based on sequencing with long reads.

As explained in our reply to your comment #4 (see above), we have discussed this directly with an author from Namasivayam et al. (2021) and have adjusted all our statements to adequately reference their work.

14. Line 177: "The length of mitochondrial reads ranged from 87 nt to 17,424 nt"It would be beneficial to include a histogram depicting the length distribution of the obtained reads. It's worth noting that nanopore reads tend to be shorter than Illumina reads

See point 5, we included a supplemental table (Supplementary file 2) showing read length distribution of mitochondrial ONT reads.

15. Line 194-195 "we found that only a small fraction of all possible block combinations areprevalent within the genome" this has been previously described (PMC8092004)

We added this to the sentence in question (line 200-202):

“[…] confirming block combinations observed previously (Namasivayam et al. 2021) and adding combination frequencies based on higher read numbers.”

Please see also our answer to your point #4.

Line 201. "This indicates that the genome's flexibility is limited and that not all block combinations are realized". This is consistent with the findings published by Namasivayam et al. in 2021, which have already established that the combination of the 21 blocks is non-random.

We added the citation to this statement: ”[…] (see also Namasivayam et al. 2021).”

16. Line 205: "All combinations are well covered in our ONT results and helped to refineblock borders relative to previous annotations (Figure S2)" In the supplementary materials the authors say: "However, the blocks Fp, Kp, and Mp frequently occur separately in the mitochondrial genome We therefore treated Fp, Kp and Mp as separate blocks and have shortened the blocks F, K and M accordingly".As far as I understand, for this very reason, Namasivayam and collaborators annotate them as partial fragments, which may appear in other regions but are, in turn, parts of larger F, K, and M fragments. To redefine the segments F, K, and M without the sequences corresponding to Fp, Kp, and Mp, as shown in Figure S2, these fragments should be distinct from the 'partials.' In other words, segments of the type (F minus Fp), (K minus Kp), and (M minus Mp) should appear in the reads, and should be distinguishable from Fp, Kp, and Mp. If this distinction is made, I am satisfied with the new definition. However, if such a separation is not evident, it seems important to clarify it in the text or to reconsider this new definition.

Implied in the concept of sequence blocks are active recombination sites. Thus, the occurrence of Fp, Kp and Mp in other combinations than with (F minus Fp), (K minus Kp) and (M minus Mp) requires their definition as single, distinct blocks. Consequently, the remaining sequences (F minus Fp), (K minus Kp) and (M minus Mp) are also blocks, on the other site of the recombination point. This is similar to many other blocks in the genome. For example, in the case of blocks P and I, block I also does not occur in other combinations than with block P in the genome whereas block P does (in combination with block H). This justifies having blocks P and I as separate blocks, rather than having a combined block "PI" or defining a block Ip and I. In sum, our new annotations harmonize block definitions.

17. Lines 221-223: "This suggests that there is no need to postulate mechanisms of genomic or posttranscriptional block shuffling to arrive at full-length open reading frames."The authors argue that invoking mechanisms of genomic or post-transcriptional block shuffling is unnecessary to explain the presence of full-length open reading frames, given that genes represent 2-3% of mitochondrial sequences. However, there is a missing estimate regarding the probability of encountering all three genes within a single molecule or mitochondrial genome, as well as the total number of sequenced mitochondria. Consequently, the statement appears overly assertive. In the absence of alternative mechanisms for generating complete genes, this would mean that at most only 1646 mitochondrial genomes would have been sequenced.To comprehensively address this issue, the authors should consider discussing this scenario further. They should also provide information about how many reads they found containing all three genes and how many contained two of the genes.

We do not think that the three ORFs need to be on one molecule since many mitochondria genomes are fragmented. Stochastic inheritance of a multi-copy genome can work just like strict inheritance of defined chromosomes. To address this is clearly beyond the scope of our work and would need controls for mitochondrial integrity and number during preparation that we simply do not have. But the numbers shown demonstrate that there are full-length ORFs and thus that it is most parsimonious to assume that this suffices to support mitochondrial gene expression.

Regarding the exact numbers that the reviewer would like to see:

We found 18 reads containing all three ORFs and 369 containing two different ORFs at least once. If we calculate the numbers of reads containing a full-length ORF relative to all reads that are long enough to fit an ORF, we arrive at the following numbers: coxIII: 5.1%, cob: 8.6%, coxI: 11.1%.

We added these numbers to the text (line 225-227).

“Thus, of all reads long enough to carry one of the following ORFs, 5.1% contain full-length coxIII, 8.6% contain full-length cob, and 11.1% full-length cox1, respectively (Supplementary file 5).”

Furthermore, to address the reviewer’s concern, we toned down our statement on post-

transcriptional shuffling (line 227-229):

“This may be adequate for the expression of the encoded proteins; however, we cannot currently exclude the possibility of genomic or post-transcriptional block-shuffling that could lead to more complete open reading frames”

18. Lines 249-250 "using the block combinations identified here by ONT sequencing " which is the difference between blocks identified here with those on Namasivayam ? The division of M, K and F fragments?

Yes, the differences concern the mentioned blocks, but also some other block adjustments as shown in Figure 2—figure supplement 2 (adjusted sequences uploaded in GenBank accessions OR086910-OR086916). We used the adjusted blocks for mapping due to reasons explained in our answer to point #16.

19. Line 287: "The six remaining small RNA fragments are specific to *T. gondii*"I would suggest being more cautious in this sentence by stating that they were not found in other organisms. Given the similarity of the mitochondrial genome between *T. gondii*, N. caninum, and other coccidians, it would be expected to find them in these organisms as well.

We thank the reviewer for bringing up this issue. It’s true that the sequences of these small RNAs are conserved among other cyst-forming Eucoccidians. We corrected the sentence in the manuscript accordingly and changed the title of the last column in Supplementary file 7 to “only found in cyst-forming Eucoccidians”.

Corrected sentence (line 297-299):

“Six further sRNAs are exclusively conserved within cyst-forming Eucoccidians, the closest apicomplexan relatives of *T. gondii* (last column in Supplementary file 7)*.”*

20. Line 300 "Among the novel small RNAs identified, there is also a class that was only detectable due to our insights into genome block combinations."A valid strategy is to map the small RNAs to the generated nanopore reads or to an assembly made with these reads, rather than solely relying on the single blocks or combinations of blocks, as this approach would yield the same result.

This would have been an alternative strategy. An assembly of a Holo-genome is however not possible. Based on all sequencing efforts to date, we would hypothesize that the genome is represented by multiple DNA fragments. Still you are right that we could have mapped the RNA reads to the entire set of reads / read clusters. We would have had to disable selection against multi mapping reads and we would have the problem that nanopore reads have many sequencing errors. Therefore, we decided the cleaner and faster method is to map against a set of all block combinations found. The results should not differ.

21. Line 444: "Upon closer scrutiny, however, the reshuffling appears limited to specific block borders and is not random" This was already established by Namasivayam et al. 2021.I would like to highlight the potential for a more comprehensive examination of the mitochondrial genome in the discussion. While the proposed explanations for the presence of sRNAs at the 'block borders' appear plausible, it's worth noting that the definition of these blocks is artificial rather than biological. I think it is interesting to discuss without the concept of block sequences, but of sequences existing in the mitochondrial genome. Therefore, it's important to discuss whether these sequences (the block borders) are consistently present in all mitochondrial genomes. The total cumulative length of the blocks is 5.9 Kb, which is relatively small and comparable to one of the smallest mitochondrial genomes on record. It is conceivable that recombination and the generation of new sequences play a role in expanding genomic space for encoding, such as RNAs.

The blocks are defined by recombination events that are implied by sequence combinations found. Therefore, the block definition by Namasivayam et al. 2021 is based on biological processes and not arbitrary / artificial. We agree however with the reviewer that recombination increases the genomic space, in particular when comparing with mitogenomes from relatives like *Plasmodium* species. To account for this interesting idea, we added the following sentence to the discussion:

line 592-595:

“In fact, the sequences at recombination sites could be regarded as an expansion of the mitochondrial genome sequence space, which is not available to other Apicomplexans like the genus *Plasmodium*.”

22. Line 535-536 "We developed a protocol to enrich *T. gondii* mitochondria and used Nanopore sequencing to comprehensively map the genome with its repeated sequence blocks."I find this sentence to be somewhat assertive, especially considering that they modified an existing protocol and obtained results that may not be optimal. Additionally, they have not compared their protocol with other available methods for mitochondrial enrichment.

We agree that this was not appropriately phrased (and cited) – please see our answer to your point #1.

Some points related to Method section:23. In none of the experiments is it specified how many parasites were initially used as a starting point

We added the missing information about the number of parasites we used for organelle enrichment. For all the other experiments performed we used a specified amount of RNA or DNA rather than a certain number of parasites.

The adjusted sentence reads (line 735-739):

“Freshly lysed *T. gondii* cultures from four T175 flasks (according to 8*10^8^ parasites) were filtered through a 3 μm pore size polycarbonate filter to remove host cell debris and harvested by centrifugation at 1500 x *g* for 10 minutes.”

24. "Masking NUMTs in the *T. gondii* nuclear genome" it's unclear whether the authors utilize all hits or filter the results of BLASTN. It would be helpful if they specify the criteria for filtering, such as identity percentage or query coverage. Additionally, it's not clear how they generate the GFF3 file from the BLAST results, or whether they instead create a BED file. Providing clarification on this process would enhance the reproducibility of their methods.Moreover, it would be beneficial if the authors include information regarding the number of sequences they intend to mask, the average length of the NUMTs, and the total percentage of the genome these masked sequences represent.

We did not filter the BLASTN results given that NUMTs can be of small length and have diverged relative to their mitochondrial counterpart (Namasivayam et al. 2023).

The interval information given in the BLASTN tabular output was manually integrated into a GFF3 file format using Excel tools.

We reformulated the sentences (line 850-854):

“All hits obtained in the BLASTN tabular output were manually integrated into a GFF3 file format. NUMTs in the nuclear genome were masked with bedtools MaskFastaBed (Quinlan and Hall 2010) based on the intervals defined in the GFF3 file. In total, we masked 8118 sites with an average length of 92 nt representing in total ~1% of the nuclear genome.“

25. Line 657 "Mapping results were filtered using SAMtools"The text does not specify the filtering criteria or the parameters used for this process.

We specified the criteria and parameters as followed (line 860-861) :

“Mapping results were filtered using SAMtools view (Version 1.15.1) (Li et al. 2009) (-f read is unmapped).”

26. Line 673 establish "No matching reads were found" in the "Sequence comparisons of ONT reads found here with published ONT reads for the *T. gondii* mitochondrial genome" but in the results the authors say: "While smaller reads of our dataset are found in full within longer reads in the published datasets, we do not find any examples for reads that would be full matches between the dataset.Could you provide a more detailed explanation? Specifically, I would like to know how many reads from the dataset (including their length) are also present in other datasets, and at what minimum length do they cease to coincide?

We apologize for the contradiction. We deleted the sentence "No matching reads were found" from the method section, which was meant to describe that we do not find identical reads.

We found 1555 reads of our dataset being entirely part (defined as ≥99% read identity) of reads from the Namasivayam et al. dataset and 212 reads of their dataset being fully found as part of reads from our dataset. The longest reads that coincide are around ~2500 nt. The sequence of a 2521 nt long read from our dataset was found completely as part of a 5574 nt long read from the Namasivayam et al. dataset. Conversely, a 2479 nt long read from the Namasivayam et al. dataset was entirely found within a 4149 nt long read of our dataset.

We added the following sentence to the methods section (line 877-879):

“We found 1555 reads of our dataset being entirely part (defined as ≥99% read identity) of reads from the Namasivayam et al. (2021) and 212 reads of their dataset being fully found within reads from our dataset.“

27. 689 – The text does not specify the filtering criteria or the parameters used for Samtools filtering process.

We added the missing specifications as followed (line 897-898):

“Mapping results were filtered using SAMtools view (Version 1.15.1) (Li et al. 2009) (-f read is unmapped).”

28. Lines 689-693 Please describe better the methodology used.

We had used default settings in Bowtie2 and added this info (line 898-901):

“Unmapped reads were kept and mapped with Bowtie2 using default settings against the *T. gondii* mitochondrial genome using all combinations of the sequence blocks found in our genome sequence analysis (GenBank accession MN077088.1 – MN077111.1, OR086910 – OR086916) as determined in our ONT DNA sequencing data.”

29. Line 696: the program is fastp not fastq (Chen et al. 2018)

Thanks, we corrected it.

30. Line 697: what do you mean only the ends of the reads were mapped? how many bases? Or do they mean that they map the reads fowrards and reverse reads?

The sentence is misleading, we used a tool that calculates for each base in the reference the number of mapped reads that have their 3’ and 5’ end at this specific position. We corrected the sentence as follows (line 905-907).

“The number of mapped 3’ and 5’ read ends at each position in the reference was calculated using Bedtools Genome Coverage 2.30.0 (Quinlan and Hall 2010).”

SignificanceIn this article, the authors delve into an intriguing topic, aiming to enhance our understanding of the organization of the mitochondrial genome of *T. gondii*, a parasite of significant importance in both human and animal health contexts.In essence, their approach involves enriching mitochondrial material, followed by genome sequencing and the analysis of mitochondrial short RNAs. They achieve a remarkable depth of mitochondrial sequencing and generate valuable RNA data. Furthermore, their efforts lead to the discovery and annotation of new short RNAs.Overall, the article is well-crafted and presents compelling results. However, it's worth noting that, at times, the authors appear somewhat self-congratulatory, and certain results might be perceived as overly ambitious. Nevertheless, the discussion is aptly constructed.Reviewer #3

We are immensely grateful for the detailed and thoughtful review you provided, particularly your insightful comments on our analysis of ribosomes, which we found to be highly valuable for enhancing the quality of our work. As you will see, we have added results from several experiments to this manuscript in response to your suggestions.

Evidence, reproducibility and claritySummaryIn their manuscript, Tetzlaff et al. report a substantially improved protocol for the isolation of mitochondria from the parasitic apicomplexan *Toxoplasma gondii*, which allowed improved sequencing and in-depth analyses of the organism's peculiarly complex mitochondrial genome. Follow-up small RNA-sequencing made it then possible to confirm the expression of fragmented mitochondrial ribosomal RNAs (mt-rRNAs) and to identify a dozen new RNA species of unknown function. The authors document not only multiple *Toxoplasma* mitochondrial genes that overlap one another-including rRNA and protein-coding genes, otherwise a rare occurrence-but also show that some fragmented rRNA genes recombine, effectively leading to multifunctional sequence segments, another rare feature and consequence of the peculiar architecture of the organism's mitochondrial genome. Lastly, the authors confirm that products of three genes presumed to encode pieces of the highly fragmented mitochondrial large subunit (mtLSU) rRNA do indeed assemble-presumably with additional components-into large molecular-weight complex(es).Major commentsKey conclusions of the manuscript are that *Toxoplasma*'s mitogenome encodes overlapping rRNA and protein-coding genes, divergent and chimeric rRNA pieces, and several small RNAs (sRNAs) of unknown function. Provided evidence is very solid for certain aspects of the study, but objectionable for the others as detailed below.1. The extent of the presented analysis of rRNAs and unassigned sRNAs seems lacking. In several places of the manuscript, the authors wonder about potential implications of divergent rRNA sequences, but their analyses appear to have been limited to sequence similarity searches. Had modelling of secondary structure interactions been attempted, this conundrum could potentially be solved. Importantly, similarity searches (to conventional rRNAs) were performed using BLASTN, which is a rather crude tool for the purpose, instead of covariance models/HMMs. It is therefore not entirely surprising that some sRNAs remained unassigned. Admittedly, recognizing rRNA motifs in divergent RNAs is a challenging issue. However, it is important to not conflate similarity to conventional rRNA and the molecule's functionality as an rRNA, i.e., sequence divergence does not necessarily disqualify the unassigned sRNAs as potential rRNAs. Mitochondrial rRNA sequences are among the most divergent, often constrained only by base-pairing, if at all, as has shown the research on kinetoplastid and diplonemid mt-rRNAs, which contain very few conserved elements and very few base pairs (e.g., Ramrath,2018,Science & Valach,2023,NAR). Even in generally less divergent cases such as green algae, the fragment encoding a highly divergent and derived 5S-like rRNA has only been recognized as such only after the mitoribosome structures were determined (Waltz,2021,Nature Comm & Tobiasson,2022,Nature Comm). It would not be surprising if the same was the case for *Toxoplasma*'s fairly quickly evolving mitochondrial genome.

We certainly did not want to imply that lack of sequence similarity rules out inclusion into a ribosome. Regarding structural analyses and more refined tools to identify similarities to rRNAs: we agree with the reviewer that such bioinformatic tools can only give tentative hints, no answers. We therefore opted for providing experimental evidence that several of the unassigned small RNAs are found in ribosomes (see extended Figure 6, new figures 7 and 8). We furthermore modified a statement on sRNA:rRNA sequence similarity and incorporated the citations suggested by the reviewer:

Line 339-343:

“None of the three RNAs had detectable homologies to rRNA based on simple sequence searches, but structural conservation cannot be ruled out. Mitochondrial rRNA sequences from kinetoplastids and diplonemids show very little sequence conservation but are still part of the mitoribosome (Valach et al. 2023; Ramrath et al. 2018), suggesting that future analyses might uncover hidden rRNA similarities.”

2. The discovery of overlapping protein-coding and rRNA genes is intriguing, but the authors do not explain why it should be considered as fundamentally groundbreaking as the 'Abstract' and 'Discussion' make it sound. Gene overlaps are found in mitochondria of many organisms (e.g., fungi, animals, various protists), especially of tRNA and protein-coding genes. Even in *Plasmodium*, a rather close relative of *Toxoplasma* studied in the presented work, LSUB (rRNA) gene overlaps cob (protein) gene in the antisense orientation. Admittedly, the extent of the overlaps in *Toxoplasma* does seem fairly high at a first glance, but it is necessary to provide more data and, importantly, broader context to make the case that *Toxoplasma* overlaps are in any way special. For instance, what is the average size of the overlaps? What is their cumulative size? How does their extent (i.e., the size of overlapping coding sequences compared to the total length of coding sequences) compare to gene overlaps in other (mitochondrial) genomes?Certain additional aspects of the analysis and interpretation of protein- and/or rRNA-coding sequence overlaps are somewhat underdeveloped. For example, are the RNA-coding regions that overlap protein-coding sequences more divergent in those three conserved proteins compared to other organisms, i.e., does their function as rRNA take precedence, or is the converse the truth, i.e., are the rRNA sections more divergent? RNA19 (overlapping coxIII and cob) is the only example discussed in depth, but at least a short sentence summarizing the overall picture would be useful. As for the authors' interpretations, proposed formation of sRNA:mRNA hybrids, through which sRNAs could by implicated in facilitating mRNA recognition by the mitoribosome, is an interesting hypothesis, but a simpler scenario, which is given very little space, is that the genes happen to overlap by chance and that the overlaps are merely a consequence of genome compaction (a phenomenon that the authors rightly highlight). Without a comprehensive analysis, it is impossible to conclude which possibility is more likely. For instance, if both protein-coding and non-protein-coding sequences are divergent, this would indicate that there are few evolutionary constraints and so the fact that these sequences overlap means very little and might be just due to neutral drift, an effect of genome compaction without much consequence for the organism.Lastly, considerable significance is attributed in the study to the presence of antisense overlaps, especially between rRNA- (or sRNA-) and protein-coding genes. Yet, the overall extent of sense and antisense overlaps in the *Toxoplasma* mitogenome is quite similar, which-again-seems to point to a neutral evolutionary process. Can the authors elaborate if this aspect of the genome architecture was taken into account and if they regard it as of lesser relevance (and why, if so)?

We are at a loss why the reviewer thinks that we mark the mRNA-sRNA overlap as “groundbreaking”. This is not our wording and this is not our main finding. In the abstract, we write “we find that many small RNAs originated from the junction sites between protein-coding blocks and rRNA sequence blocks.“ In the discussion we write: “A few cases of mRNA sequences overlapping in antisense orientation with rRNA have been described in mammals and yeast (Kermekchiev and Ivanova 2001; Coelho et al. 2002). Sequences homologous to rRNAs have been found in many coding regions in sense and antisense orientation (Mauro and Edelman 1997). It has been suggested that this could link ribosome production to other cellular processes by reciprocal inhibition of mRNA and rRNA expression (Coelho et al. 2002). It is possible that *RNA19* and ribosome production could be balanced with *cob* protein production by tuning the processing or RNA degradation of *cob* mRNA.” We honestly do not feel that we overstated anything here and in fact we cite prior ideas on this phenomenon.

What we think is outstanding and novel is that there are small RNAs made at recombination sites and that these are functional – incorporated into ribosomes. We show this in our revision for several more sRNA examples in response to the helpful remarks by this reviewer – see below. The further news is that the block border sRNAs fuse mRNA and non-mRNA sequence, which is discussed in the snippet above for RNA19. Maybe we are ignorant and missed something, but where has it been demonstrated that a short RNA from a genomic recombination site that fuses coding and noncoding sequences is incorporated into ribosomes?

In sum, we do not think that extending the manuscript by analyzing the overlaps in greater detail adds anything to our prime conclusion: sRNAs at block borders are incorporated into ribosomes. But for the reviewers’ sake: there are 9 sites in sRNAs with overlap to mRNAs and other sRNAs, average size of overlaps: 20 nt, cumulative size overlaps: 176 nt, longest overlap: 49 nt (RNA1/2), shortest: 8 nt (RNA17/13). We doubt however that these numbers are interesting for the readers and would rather not include them in the manuscript.

3. Another controversial issue concerns prevalent sequence block combinations and their impact on mitochondrial gene expression regulation. The authors postulate that 5′-terminal blocks of protein-coding genes always occurring near other protein-coding blocks has some functional significance. However, concluding this from just two cases (even if out of two) is quite speculative and seems like reading too much into a pattern that could very well be due to chance alone. The authors argue that the fact that 5′ ends of coxI & coxIII genes overlap is another indication of potential gene expression coordination. While it is possible to envisage such a regulation because of the 5′ termini proximity, the overlap between these genes means that their connection is hardwired into the genome, making it difficult to compare this particular case to the other sequence blocks. Arguably, it is tempting to speculate that an evolutionary pressure exists to coordinate protein expression and such a coordination does not indeed seem implausible, but the presented data and arguments are not convincing. The authors should at least expand on their ideas in the 'Discussion' and indicate potential experiments and/or which additional data could support (or refute) their speculation.

We agree that this is a speculation and that we would need to suggest an experiment to address this. Possibly, this could be tested by mutagenizing the linker regions between coding regions. e.g. with TALEN-based genome editors, and monitoring whether there are joined effects on the expression of adjacent genes. However, this is not established for Apicomplexan mitochondria. So, on second thought, we deleted this paragraph from the discussion since we really do not want to distract from our main findings on sRNAs.

4. My last major point concerns the experimental examination of large-molecular weight complexes and the interpretation of its results. To prove incorporation of the sRNAs into the mitoribosome, i.e., confirm that they do indeed represent rRNAs, the authors opted to investigate their distribution across a sucrose velocity gradient. This is a relatively simple and powerful approach and although it does not provide an irrevocable proof, it can be used to gain very useful insights. However, the presented design has critical flaws: 1) all sRNAs selected for Northern blot were mtLSU components, so only the mtLSU would be detected;

We now included two SSU rRNAs

2) a single cytosolic LSU component was used as the control, so the distribution of cyto-SSU subunit, cyto-ribosome, and cyto-polysomes is actually unclear;

We now purified organelles including a benzonase step, which removes the cytosolic rRNAs. Besides, we focused on the mitochondrial ribosomes and can draw conclusions on their positioning based on added controls (see below) in the absence of information on cytosolic ribosomes.

3) the authors' interpretation relies on the assumption that both mitochondrial and cytosolic ribosomes preserve their association as polysomes, but no relevant control is provided for this. For example, in Figure 6, fractions 6-14 clearly contain cyto-LSU, but polysomes (e.g., disomes) might just as well start in fractions 12-14; without additional controls, or at least continuous monitoring of UV absorbance across the gradient (to show a typical polysomal pattern), it is not guaranteed that what was detected actually included cyto-polysomes.

We now analyzed a total of seven RNAs under ribosome-dissociative conditions and see a strong shift in the sRNA signals towards lower molecular weight fractions.

The main concern, however, is the migration of mitoribosomes. First, the authors presume that the fractions 7-8 contain the mitochondrial monosomes because they are the fractions closest to the gradient top. This is not guaranteed. In fact, based on the experience of our and our colleagues' labs and taking into consideration the conditions used for the described experiment (more precisely, the use of Triton and deoxycholate, which in many organisms lead to mitoribosome subunit dissociation), it seems quite likely that fractions 7-9 actually contain separated mtLSU, not monosomes. Fractions in higher sucrose concentration would then represent monosomes and possibly assembly intermediates, though perhaps also a minor polysomal fraction (if the interactions are preserved in the conditions used). In particular, if the assembly process in Apicomplexa is as complex as in Euglenozoa (e.g., see papers on kinetoplastid mitoribosomes Saurer,2019,Science & Tobiasson,2021,EMBO Journal), which does not seem unlikely in *Toxoplasma* given the necessity to incorporate ~15 distinct rRNA pieces per mitoribosomal subunit, then the assembly intermediates might form ribonucleoprotein complexes that migrate quite far into a sucrose gradient (e.g., as in kinetoplastid mtSSU, Maslov,2007,Mol Biol Parasit). Thus, while it can be reasonably well argued that the detected RNAs co-migrate with the mtLSU (and possibly mito-monosome), the claim that they associate with mito-polysomes is open to question. More critically, investigating only sRNAs that are clearly identifiable as rRNA pieces-and all from the mtLSU at that-does not automatically prove that all sRNAs associate with the mitoribosome.To argue that the unassigned sRNAs are associated with mitoribosomes, northern blots of as many as possible (but at the very least one) unassigned sRNAs are absolutely necessary. However, I encourage the authors to consider performing additional experiments to address the issues raised in the preceding paragraph: for example, a western blot of mitochondrial ribosomal protein(s), a northern blot with at least one mtSSU rRNA fragment (since all three shown are from mtLSU), as well as a test that would examine the influence of detergents on mitoribosome stability (e.g., use milder detergents such as digitonin or dodecylmaltoside). Furthermore, if experimental conditions are identified allowing subunit dissociation, it would be possible to discern to which subunit which sRNA belongs and, importantly, whether the unassigned sRNAs are just "disguised" rRNAs (simplest explanation) or something completely different (speculative explanation seemingly favoured by the authors). All this would substantially boost the significance of the presented work.

We thank the reviewer for these thoughtful comments and the many helpful suggestions. Following the reviewer's advice, we have opted for analyzing a total of seven sRNAs, including two RNAs assigned to the small subunit and one RNA not described to share homology with *E. coli* rRNA. Importantly, we included an analysis using standard dissociative conditions (10mM EDTA, no Mg, 300 mM K) for all sRNAs. Finally, we also include an analysis of the three mRNAs and one mitoribosomal protein. We have furthermore used mitochondrial preparations as starting material, which has the advantage of fewer cross-hybridizations and better detectability of sRNAs in general.

This demonstrated:

High molecular weight complexes deep in the gradient are sensitive to EDTA treatment. Most parsimoniously, these are polysomes, since they comigrate with mitochondrial mRNA and a ribosomal protein we tested (L11:HA, required making of a transgenic line) , which are equally EDTA-sensitive.SSU and LSU sRNAs show a distinct distribution in the gradient, based on which we could assign hitherto unassigned RNA29 to LSU.

Here are some answers to further reviewer comments:

There are no antibodies available that would recognize mitochondrial ribosomal proteins, but we included now analyses of SSU sRNAs and tagged a ribosomal protein (L11) to allow gradient analysis.We showed that the high-molecular weight complexes containing sRNAs / rRNAs / L11 are sensitive to EDTA treatment, suggesting they are indeed ribosomes / polysomes.We did not intend to favor other hypothesis for sRNA functions over them being rRNA fragments. We hope this is now more obvious giving the longer discussion of our gradient analysis. Still, we think it is important to consider other explanations for their accumulation.

The massive addition of data (extension of figure 6 and new figures 7 and 8) led to various textual changes in the last chapter of the results as well as in the last chapter of the discussion. There are so many changes that we refrain from listing them all here and have to refer the reviewer to the manuscript itself.

General commentsThe word "novel" is rather overused in the manuscript. At several places, it is inappropriate, as the presented results are not as unprecedented as the manuscript makes them sound; at other places, it might be acceptable, but as the word's meaning is vague, the text would benefit from using more informative term(s) instead. The former case is exemplified by the sentence at the lane 102 "Here, we present a novel method for enriching organellar nucleic acids" – "novel" does not simply mean "new", but alludes to "unprecedented"; yet, the devised method, albeit clever, is a modification of existing approaches. The sentence at the lane 182 illustrates the latter case where "novel blocks" are mentioned, but "previously not detected blocks" would be more appropriate and to the point.The labelling of 5′ and 3′ is inconsistent throughout the manuscript – sometimes the prime is used, sometimes the apostrophe, sometimes it is the single quotation mark.

We removed the “novel” in “novel method” in response to a comment by reviewer #2.

We went with the reviewer’s suggestion to rename “novel blocks” to “previously not detected blocks” or “additional blocks” and also applied this to “novel transcripts” and “novel rRNAs”. The word “novel” is no longer found in our text; only in the references.

AbstractIn light of the raised concerns, the authors should consider carefully rewording this section, as some of the formulations are mis-representing the data and lead to unjustified generalizations.

We rephrased the section on block arrangements. Since we strengthened our conclusion on functional sRNAs (as rRNAs) with further experiments, we uphold the part that refers to RNAs.

Introductionlanes 72-73: "How rRNA fragments are assembled into functional ribosomes remains an enigma." – Without proper context, this statement feels like an exaggeration. Fragmented rRNAs are known from other species and their mitoribosome structures were determined in the past few years (i.e., Tetrahymena, Polytomella, Chlamydomonas). Arguably, these mt-rRNAs are not as fragmented as in *Toxoplasma*, but at the very least, it is clear that base-pairing of rRNA pieces and RNA-binding proteins play significant roles in the process. If the authors think that this is not the case in apicomplexans, this should be at least alluded to, if not explained.

OK, the absoluteness of that statement is indeed misleading. We changed it to:

Line 74-75:

“How the multitude of rRNA fragments are assembled into functional mitoribosomes in *T. gondii* remains unknown.”

l. 80-83: The paragraph mixes information on *Plasmodium* and *Toxoplasma*. To a non-initiated reader, this can be quite confusing. It would be useful to specify which species the authors refer to.

We added species information:

Line 82-86:

“In addition, the depletion of nuclear encoded mito-ribosomal proteins of *T. gondii* (Lacombe et al. 2019; Shikha et al. 2022) and *P. falciparum* (Ke et al. 2018) led to defects in the assembly of ETC complexes and in parasite proliferation, suggesting that mito translation is important for parasite survival.”

l. 83-86: The information on the atovaquone impact lacks reference(s).

Thanks, we added the appropriate references as followed.

Line 86-89:

“Resistance to the antimalarial drug atovaquone in *P. falciparum* and *T. gondii* has been linked to mutations in the *cob* (*cytB*) gene of the mitochondria (McFadden et al. 2000; Syafruddin et al. 1999; Srivastava et al. 1999), further supporting the idea of active, essential translation in apicomplexan mitochondria.”

l. 105: "demonstrated that they are incorporated into polysomes" – In light of the issues raised above and if the authors opt not to expand the work as suggested above, this claim (and similar throughout the text) should be emended.

We provide additional information that this is polysome based – by adding an analysis of sRNAs, mRNAs and L11 distribution under ribosome-destabilizing conditions. Please see the last chapter of the Results section.

l. 106-108: "allowed us to identify novel transcripts, many of which originate from block boundaries and contain mixed origins from coding and noncoding regions." – This sentence would benefit from rephrasing because it is difficult to comprehend (the sequences overlap protein-coding and non-protein-coding regions, but do not contain any origins).

We rephrased (line 108-111):

“The combination of DNA sequencing results and transcriptome analysis also allowed us to identify previously undetected transcripts, many of which originate from block boundaries and represent fusions of coding and noncoding regions.”

Resultsl. 115-117: "cell fractionation method that takes advantage of the differential cholesterol content in plasma membranes" – Does *Toxoplasma* contain cholesterol? Perhaps it might be more practical to refer to sterols (since the effect of digitonin is not limited to cholesterol).

To be more precise we have replaced “cholesterol” with "sterol".

Line 118-122:

“We modified a cell fractionation method that takes advantage of the differential sterol content in plasma membranes and organellar membranes (Esseiva et al., 2004; Subczynski et al., 2017). We incubated cells with the detergent digitonin, which selectively permeabilizes sterol-rich membranes, leaving mitochondria and other organelles intact.”

According to literature, *T. gondii* cannot synthesize sterols, but instead scavenges cholesterol from the host cell (Coppens et al. 2000). There is a lot of evidence supporting cholesterol incorporation into *T. gondii* membranes (Foussard et al. 1991, Coppens et al. 2000).

l. 147: "significant increase" – It might be useful to specify that the increase was ~42-fold, so that readers can see the extent of improvement; it has the advantage of really highlighting the achievement.

We value the reviewer's suggestion on how to better emphasize the achievement. We have integrated the fold-change into the manuscript.

Line 177-179:

“This is a 42-fold increase in the sequencing depth of the *T. gondii* mitochondrial genome compared to previous attempts (Supplementary file 1), which can be attributed to the effectiveness of the purification process.”

l. 180: "have been lettered from A-V" – Rewording to "designated by letters from A to V" works better.

We followed the Reviewer’s recommendation and rephrased the sentence.

l. 213-218: This section is essentially a discussion so should be moved the corresponding section of the manuscript.

Since the reviewer pointed out that our suggestion is a speculation and we in response deleted the corresponding discussion, we shortened this part as well.

Line 220-221:

“[…] it is conspicuous that their three 5'-ends are always flanked by blocks also encoding a protein (Figure 2C), although it is unclear whether this is of functional relevance.”

l. 262-265: cotranscripts/transcript isoforms – It is a matter of nomenclature, but it seems more appropriate to refer to "a transcript containing LSUF and LSUG regions" instead of a co-transcript, because in the latter case, one then expects that these two will be separated in a following processing step, which-as the authors demonstrate-is clearly not the case for the vast majority of the population of these rRNA pieces. Given the prevalence of the larger pieces, it seems more appropriate to refer to the "smaller transcript isoforms" as possible degradation products and not isoforms, which implies some kind of functional relevance.

Yes, we agree that this makes more sense. Here is the revised sentence:

Line 274-278:

“Our sequencing results suggest that there is accumulation of transcripts containing LSUF and LSUG regions and LSUD and LSUE regions, respectively. Both of these transcripts were verified via northern blotting (Figure 3—figure supplement 1). The longer transcripts were found to be much more abundant than smaller transcripts that were also detected, suggesting that the longer transcripts represent the functional rRNA fragments in *T. gondii* mitochondria.”

l. 281: In the section "Discovery of novel rRNA fragments", it might be useful to provide a graphical representation or at least a sentence summarizing all different categories of sRNAs. For instance, what is missing from the text is that there are 11 species for which homologous sequences in "conventional" rRNAs were not identified and out of these only 4 seem to have sequence homologs in other Apicomplexa. In addition, in Supplementary file 5, the authors could indicate where these homologs are located in *Plasmodium*, since these appear to be newly identified candidates for *Plasmodium* sRNA species/rRNA pieces.

We tried to describe short RNA categories and made a table summarizing the categories (Supplementary file 8):

Line 294-304:

“Out of the 34 small RNA fragments identified, 23 have not been described previously for *T. gondii* (see Supplementary file 7 and Figure 3—figure supplement 2). 17 of the 23 are homologs of sRNA fragments in the apicomplexans *P. falciparum* and *E. leuckarti* (marked in bold in Supplementary file 7). These were named according to their *Plasmodium* homologs (Feagin et al. 2012). Six further sRNAs are exclusively conserved within cyst-forming Eucoccidians, the closest apicomplexan relatives of *T. gondii* (last column in Supplementary file 7). We assigned numbers to these sRNAs extending the *Plasmodium* nomenclature (Feagin et al. 2012). We next asked, how many of all 34 sRNAs have homologies to rRNA from *E. coli*. We could find twelve LSU homologs and eleven SSU homologs in accordance with previous analyses in *P. falciparum* (Feagin et al. 2012; Hillebrand et al. 2018 ; see Supplementary file 8). Only for the *P. falciparum* LSU fragment LSUC and the SSU fragment RNA12, we were unable to identify corresponding homologs in *T. gondii*.”

l. 313-314: "In general, block combinations lead to the expression of novel RNAs in *T. gondii* that are not found in apicomplexan species with a simpler genome organization. " – It is not clear where this generalization comes from: Figure S5A shows that RNA5, RNA7, RNA23t extend across block borders (but based on Table S5 are not unique to *Toxoplasma*), while only RNA31 and RNA34 are both absent from other Apicomplexa and extend across block borders – yet, this is still less than half of all newly identified sRNAs. In addition, the novelty claim is not clear either: based on the presented data, several sRNAs that overlap are clearly present in other apicomplexans (e.g., RNA1 and RNA2) and thus are not completely new, but merely more divergent in *Toxoplasma*, because parts of their sequence have been replaced by the shared sequence segment.

We think that this is poor wording on our side. What we wanted to say is that there are sRNA in *T. gondii* that are not found in *Plasmodium* and that these are at block borders. Such RNAs are highlighted in Supplementary file 5 (now Supplementary file 7) in the last column. We rephrased this statement accordingly:

Line 331-333:

“In conclusion, block combinations can lead to the expression of novel RNAs in *T. gondii* that are not found in apicomplexan species with a simpler genome organization (Supplementary file 7)”

l. 319-320: "None of the three RNAs had detectable homologies to rRNA." – Specify to which rRNAs were the sequences compared to make the inference.

RNA16 and RNA23t are homologs of *Plasmodium* sRNAs that have not been assigned to ribosomal RNA regions previously (Feagin et al. 2012). In their study, Feagin et al. employed tools which also consider secondary structure and base pairing probability of RNAs to assign these sRNAs to the ribosome. The sequence of RNA34 is conserved in *Eimeria*, however, the corresponding region is not annotated in publicly available *Eimeria* mitochondrial genomes. Using BLASTN we could not identify homology to *E. coli* rRNA regions for RNA34.

We added (line 339-340):

“None of the three RNAs had detectable homologies to *E. coli* rRNA based on simple sequence searches, but structural conservation cannot be ruled out.”

l. 320-321: "For all five coding-noncoding RNAs, homologs are present in the mitochondrial genome of *P. falciparum*." – Does this mean that they remain unassigned in *Plasmodium* as well or that they have not been previously recognized in *Plasmodium*?

We rephrased this to make clear that they have been recognized before.

Line 343-346:

“With the exception of RNA34, the RNAs representing a fusion of non-protein-coding regions and protein-coding regions, have homologous sequences in the mitochondrial genome of *P. falciparum* (Namasivayam et al. 2021).”

Confusingly, RNA34 is labeled as not having homologs in Apicomplexa in Table S5.

RNA34 does have a homologue in Eucoccidians and the sequence is also found in Eimeria, but not in *Plasmodium* (see above); we corrected this in Supplementary file 5 (now Supplementary file 7).

In addition, mentioning "coding-noncoding RNAs" is somewhat misleading because some of the sRNAs clearly code for mt-rRNA pieces.

It is debatable whether “coding” should be used for DNA sequences expressing RNAs that do not code for proteins. “Coding” to us implies the genetic code, i.e. protein coding. Nevertheless, to make this clearer, we rephrased this:

Line 336:

“[…] we also found five RNAs that combine sequences from coding and non**-**protein-coding blocks […]”

Line 343-346:

“With the exception of RNA34, the RNAs representing a fusion of non-protein-coding regions and protein-coding regions, have homologous sequences in the mitochondrial genome of *P. falciparum* (Namasivayam et al. 2021).”

l. 335-338: This section contains contradictory statements that should be reformulated. A couple of sentences prior, the authors experimentally determined that RNA19 actually overlaps only a single protein-coding sequence (coxI), but then refer to the original and demonstrably incorrect annotation of RNA19 overlapping also the cob gene.

We apologize for the contradiction. The RACE experiment was performed during the writing process and we unfortunately overlooked this sentence when revising our statements about the RNA19 overlap. We have now corrected the sentence.

line 358-360:

“This sequence similarity to rRNA is maintained in *T. gondii* (Figure 5E), which suggests that despite overlapping with coxIII coding sequence at the J-E block border, RNA19 is functional.”

l. 341: The authors mention similarity to rRNA, but do not specify which rRNA. Referring to similarity to known or conserved rRNA sequences or segments would work better. Still, the region of the block S (i.e., 5′ proximal segment of RNA19) falls into the region between helices H51 and H60 of the domain III in the LSU secondary structure, which is sequence-wise relatively poorly conserved-especially in mitochondrial rRNAs-so sequence divergence is not unexpected.

Thank you. We added this idea to the text.

Line 366-367:

“The similarity to *E. coli* 23S rRNA is, however, restricted to the noncoding block R (Figure 5—figure supplement 1).”

Line 369-371:

“It is noteworthy that the 5’ end of RNA3 is located within a region of the 23S rRNA secondary structure that exhibits low conservation (Feagin et al. 2012), which may suggest that its overlap with coding regions is potentially tolerable.”

l. 366: "Note that RNA1 and RNA2 are registered according to their shared sequence" – Unclear what "registered" means here.

We replaced “registered” with “aligned”.

l. 416-421: Specifying when reference is made to cytosolic vs. mitochondrial monosomes and polysomes would make this section and the related parts of the 'Discussion' clearer. Also, the authors clearly state here that there might be technical reasons for what they observed, but ignore this possibility in the 'Discussion' and assume that they did indeed separate polysomes.

We added data to this section and did a complete rewrite. We are speaking almost exclusively about mitochondrial monosomes and polysomes and were in the revised version careful to state when cytosolic ribosomes were meant.

Discussionl. 444: "the reshuffling appears limited to specific block borders and is not random" – How many biological replicates of nanopore sequencing were performed? Did the authors test other *T. gondii* strains? What about other apicomplexan species? Unless this has been done, there is no demonstration that the block order and block-joining frequencies documented here are (more or less) constant and that block order is under some kind of purifying selection. Hence, the conclusion that the block borders are not random is debatable. Arguably, it is not random in this particular experiment, but neither is it limited to specific blocks because most combinations have been detected (even if at low frequency; Figure S1).

After revisiting the analysis and excluding falsely identified block combinations separated by more than 10 un-annotated nucleotides, we arrive at a much clearer distribution of block combinations. This distribution is evidently non-random, confined to specific blocks. As depicted in Figure 2—figure supplement 1A, only a minority of block combinations appears more than 50 times in our dataset. Supplementary file 4, encompassing both infrequently occurring and potentially falsely identified combinations, further substantiates the notion that block combinations are not randomly distributed.

We also would like to stress that the two other attempts at sequencing the mitogenome of *T. gondii* found the same block combinations, although the composition of individual reads differed (Namasivayam et al. 2021; Berna et al. 2021). Thus, block combinations are not random.

A straightforward thought experiment underscores this point: in the case of random combinations, all 24 blocks would theoretically pair head-to-head, head-to-tail, and tail-to-tail with every other block, resulting in (24*2)^2^ = 2304 unique two-block combinations. However, even without filtering, we identified only 84 different combinations (see Supplementary file 4), with 32 of these appearing only once and 52 appearing less than 50 times in a total of 284,679 pairs.

We adjusted statements in the manuscript based on our revised analysis:

Line 202-204:

“For example, the most frequent block combination is J-B, which occurs 19,622 times in our reads. In total we identified 84 combinations of which 52 occur less than fifty times and make up less than 0.06% of the total number of combinations found (Supplementary file 4).”

Furthermore we found a mistake in the number of high-frequency block combinations found and corrected it.

Line 213-214:

“Using the 32 high-frequency block combinations we found, we generated a map centered on the protein-coding genes (Figure 2B).”

l. 450: "One intriguing finding is the obligate linkage of coding sequences" – Presuming this sentence is about protein-coding sequences, this should be reformulated because it mis-represents the actual data. Figure 2 clearly shows that protein-coding blocks are often linked to rRNA-coding blocks.

We have deleted this entire section given the comments on the low number of coding regions and thus the difficulty to draw conclusions from the few head-to-head and head-to-tail linkages of coding regions in the genome.

l. 454: "balancing the expression of coxI and coxIII" – Not clear where this information comes from, as it is not from the cited papers.

We deleted this section in response to other comments.

l. 460-461: "Our small RNA sequencing results revealed another potential advantage of the block organization of the *T. gondii* mitochondrial genome" – This should be reformulated. Clearly, the discovery of the 15 sRNAs was facilitated by the recognition of block order, but the presented argument is a bit confusing: how does the organization into blocks provide an "advantage" and what kind of advantage do the authors mean? (An evolutionary advantage or an advantage related to gene expression regulation or an advantage for their sRNA-Seq data mapping?)

The idea of an advantage was raised since the block organization can lead to the expression of two alternative RNAs and thus could increase the number of RNA species and novel evolutionary options. However, since this is difficult to understand in this introductory sentence and will be explained below, we deleted the phrase:

Line 582:

Our small RNA sequencing results revealed that 15 small RNAs span block borders.

l. 462-478: Multiple explanations are provided for the existence of sRNAs at block borders and what these sRNAs represent. While I agree that it is important to consider all options, even the more debatable ones, the authors seem to forget the simplest possibility: the identified unassigned sRNAs could well be rRNA pieces and them being encoded across block borders is not any more, nor any less surprising than the fact that protein-coding genes are encoded across (several) gene blocks.

There is a conceptual difference between a long mRNA spread across several blocks (and thus also block borders) and a biased distribution of short RNAs to block borders. The short RNAs could also have been positioned elsewhere – being short, they could all be located inside blocks, sense or antisense to coding regions or other non-coding RNAs. But almost half of all RNAs are positioned at block borders. This asks for an explanation, which we discuss here.

Regarding the idea that the sRNAs are rRNA fragments: yes, we agree and had not excluded that. We provide now experimental evidence that many sRNAs, including block-border sRNAs are part of ribosomes. We added the following sentence and discussed this in greater detail in the last chapter of the discussion.

Line 625-626:

“Regarding their function, many, if not all, of the sRNAs at block borders could be used in ribosomes as rRNA fragments, which is discussed in the next chapter.”

l. 485: "antisense RNA surveillance" – In contrast to the nuclei, the existence of a genuine antisense RNA "surveillance" mechanism in mitochondria is uncertain. Given what is known from mitochondria of other organisms (especially plants and kinetoplastids), it seems more likely that certain regions of sense and antisense transcripts are protected from exonucleases by RNA-binding proteins (RBPs such as PPR and related helix-turn-helix repeat proteins, e.g., *Toxoplasma*'s homologs HPRs discovered in *Plasmodium* [Hillebrand,2018,NAR]), leading to RNAs that partially overlap, but are actually protected from base-pairing by these RBPs. This is not taken into account in any presented explanation of the phenomenon of antisense gene overlaps.

There is no accumulation of RNA antisense to rRNAs and mRNAs. On the other hand it has been shown that long precursor RNAs exist for both strands of Apicomplexa mito genomes including *T. gondii* (work by Stuart Ralph lab). Hence, there must be degradation of antisense RNA.

We in general like the idea of RNAs as footprints, however, in Apicomplexa, most short RNAs are too long to be footprints for single HPRs or PPRs; also, there are only two PPRs described for *T. gondii*. As for HPRs, experimental links between HPR and sRNAs are missing so far. We would therefore rather not include this additional speculation in the discussion.

l. 490: "start codon. while also " – Typo: should be a comma, not a dot.

This was corrected.

l. 500: "discovery of block-border sRNAs highlights the complex regulatory mechanisms at play" – This should be reformulated: the claim is very speculative, since no hard data are provided on such regulatory mechanisms in the presented work.

We rephrased this:

Line 622-625:

“Overall, the discovery of block-border sRNAs highlights the complex biogenesis of sRNAs in *T. gondii* mitochondria and will be a starting point to understand the processing of sRNAs and their function in general.”

l. 504: "sRNAs are incorporated into polysome-size structures" – In light of the concerns raised in the preceding section, this should be reformulated.

We revised this part of the discussion completely since we provide additional data showing that sRNAs are in polysomes.

l. 539-540: The closing sentence should be reformulated. The mitogenome organization in blocks per se does not "allow" the sequences to function as both mRNA and rRNA. Rather, it seems to be a combination of 1) the compactness of the genome that seems to lead to the re-use of certain segments in both mRNA and rRNA or in two distinct rRNAs, and 2) the apparently dynamic nature of the genome (due to recombination among gene blocks) that brings together certain combinations of gene blocks.

Although we do not see why our phrasing would not be correctly representing our findings, we are happy to rephrase according to the reviewer’s suggestion:

Line 684-686:

“*T. gondii*'s dynamic block-based genome organization leads to usage of mitochondrial sequences in mRNA as well asrRNA contexts, potentially linking rRNA and mRNA expression regulation.”

Methodsl. 607: Only agarose gel separation is mentioned, but most experiments shown are of denaturing PAGE separations (which is actually mentioned in several figure legends).

Initially, the description of Denaturing PAGE was included within the chapter on agarose gel electrophoresis. We now created a separate chapter titled “Denaturing Urea-PAGE and sRNA gel blot” and added some details on the method.

Corrected version of the agarose gel electrophoresis chapter:

Line 792-794:

“RNA samples were diluted in denaturing loading buffer (Deionized formamide 62.5% (v/v), formaldehyde 1.14 M, bromophenol blue 200 μg/mL, xylene cyanole 200 μg/mL, MOPS-EDTA-sodium acetate)and separated on a 1% agarose gel containing 1.2% formaldehyde.”

added under “Denaturing Urea-PAGE and sRNA gel blot”:

Line 797-799:

“RNA was separated by denaturing Urea-PAGE (10% or 12% polyacrylamide gel for total RNA/organelle enriched RNA and RNA extracted from sucrose density gradient fractions, respectively).”

l. 636: "Paste your Materials and methods section here." – To be removed.

We have removed the sentence.

l. 662: "NUMTS" – This should be "NUMTs"; the same typo occurs at multiple places in the 'Methods' section.

We have corrected the typo in all places.

l. 704: "Homology search for novel transcript annotation" – Somewhat confusing title; it is possible to guess what the authors likely mean, but it is unclear.

We changed the chapter name to “Sequence similarity search”.

l. 715: "New block annotations can be found in GenBank." – 1) The whole community would very likely appreciate if the GenBank entries were properly annotated (i.e., genes added), not just showed sequences as is currently the case for all Namasivayam,2021, Genome Res entries (not sure about the authors' own entries because they were inaccessible). If impossible to update the entries of the Namasivayam,2021, Genome Res study, then just submitting anew properly annotated GenBank entries would be appropriate.

We are not allowed by GenBank to change the published entries, nor do we have information to add beyond what was published by the Kissinger lab. The exceptions are blocks that we redefined due to our sequencing efforts (GenBank, OR086910 – OR086916). Here, we added annotations on the sRNAs.

2) It was not possible to properly assess some of the claims in the manuscript because access to the files was not provided to reviewers, nor have been the newly submitted GenBank entries made public by the authors.

They are fully publicly available. Same for the GenBank entries.

FiguresFigure 1B – The load of total proteins into each well is unclear. Ponceau stain does not show identical loads, so it is unclear what the reader should take as the reference.

As written in the legend, 5% volume of each fraction was analyzed. The Ponceau is thus not a mass control, but shows that with increasing purification, less total protein remains, while the Western shows that mitochondrial GFP remains / is enriched.

Figure 1D -The phrasing "fragments found in the pellet fractions of the protocol" is a bit awkward. The fragments are in the pellet fractions after plasma membrane permeabilization and benzonase incubation, not in the "fractions of the protocol".

We changed the phrasing to “Both fragments are found in pellet fractions after digitonin treatment, where they are protected from benzonase digestion”.

Figure 2 – The chosen hues of red and green (for coxI and coxIII) are of such similar intensity that they are virtually indistinguishable to ~2% of the readers. A colourblind-friendly palette would be very much appreciated. For guidelines, see for example: https://www.nature.com/articles/nmeth.1618.

We thank the reviewer for the advice. We have adjusted the color palette in the schematics to make it more colorblind-friendly.

Figure 3 – The use of lowercase letters to indicate the probes (instead of the full probe names) is a nice idea and simplifies the reading experience, but the use of the same letter 'a' in different figures for different probes is confusing. Labeling each probe with a unique ID/letter and indicating this ID in the Table S6 (e.g., by adding an additional column) would work much better.

We changed the designation to unique probe IDs which have been integrated as a column in Supplementary file 6 (now Supplementary file 9).

Figure 4A – The wiggle lines for rRNAs are coloured in purple shades, which contrast with the grey colour that is assigned to them in the Figure 2. Keeping a consistent colour palette across figures would be preferable.

We now keep the color code consistent and show RNAs in Figure 2 with colored wiggle lines.

Figure 4C – If the *E. coli* sequence was on the outer lines, the *Toxoplasma* sequences could be closer to one another, which would make it easier for the reader to understand the alignment.

We changed this.

Figure 5 – Purple shades for rRNA are somewhat difficult to discern from the blue cob. Also, the 'reference' wiggles would work better if demarcated as a key because this would make it visually clearer that they are shared by the A and B panels.

We’ve revised the color scheme and also designated the legend as such.

Supplementary InformationFigure S1 – An explanation what the A and B panels show is missing.

Thanks, we have added the missing information. We now show the revised heatmap (including directionality) in Figure 2—figure supplement 1 instead of 2B and added a supplemental table (Supplementary file 4) representing the absolute numbers of occurring block combinations.

Figure S5 – It is difficult to appreciate the extent of overlaps with protein-coding sequences if these are missing from the figure (unlike in Figure 5).

The information regarding overlaps is available in various other figures; we aim to avoid further complicating this overview figure. The intent is not to display the overlaps of sRNAs.

Table S4 – Nuclear genome accession number is missing. Add "mitochondrial" to the label of the column "sequence blocks".

We added the missing information.

Table S5 – 1) It is unclear what the 'rRNA homology' refers to. (It does not seem to be the nomenclature used by Feagin et al.,2012, PLoS One.) 2) An extension of the table (or perhaps a separate table) with the cumulative size of mtLSU and mtSSU rRNA pieces, as well as unassigned sRNAs, would be useful. 3) It should also be stated somewhere if homologs of any of the rRNA pieces known from *Plasmodium* are missing in *Toxoplasma*. (If so, they could be among the newly identified short RNAs.)

We changed the column title to “assigned rRNA region” and included a footnote to clarify what we refer to.We incorporated an additional supplementary Table (Supplementary file 8) that compares fragment numbers and cumulative sizes of SSU rRNA, LSU rRNA and unassigned sRNAs between *P. falciparum* and *T. gondii*.We included a sentence in the manuscript referring to *P. falciparum* rRNA pieces that were not found in *T. gondii*.

Line 303-304:

“Only for the *P. falciparum* LSU fragment LSUC and the SSU fragment RNA12, we were unable to identify corresponding homologs in *T. gondii*.”

SignificanceSpeaking from personal experience, devising a protocol for such a substantial mitochondrial enrichment, as the study presents, is a great technical achievement, which cannot be understated, especially for a protist or any somewhat unconventional model organism. The mitoribosomal community will certainly take notice of the improved catalogue of mitochondrial rRNA pieces, while the discovery of overlapping protein-coding and rRNA genes will be of interest to those working in the field of mitochondrial evolutionary biology. The study already provides a significant upgrade from the previous attempts to understand the nature of the mitochondrial genome in *Toxoplasma* (and in Apicomplexa in general), and is well positioned to become a source of inspiration for future studies in the field. However, being at a crossroad of genomics, evolution, and molecular biology, it has certain limitations in its current form, mainly because the evolutionary and molecular biology aspects would benefit from further development (see 'Major concerns'). The text is generally well written and accompanying figures well designed, but clarifications, broader context, and less speculative interpretation would be welcome (as detailed mostly in 'Minor concerns'). To justify publication in a journal with a broad readership, the authors should provide additional experimental evidence to strengthen their case and generalize their findings.